# GDF15 controls primary cilia morphology and function thereby affecting progenitor proliferation

Katja Baur[1] , Şeydanur Şan[1,2], Gabriele Hölzl-Wenig[1], Claudia Mandl[1], Andrea Hellwig[1] , Francesca Ciccolini[1]

We recently reported that growth/differentiation factor 15 (GDF15) and its receptor GDNF family receptor alpha-like (GFRAL) are expressed in the periventricular germinal epithelium thereby regulating apical progenitor proliferation. However, the mechanisms are unknown. We now found GFRAL in primary cilia and altered cilia morphology upon GDF15 ablation. Mutant progenitors also displayed increased histone deacetylase 6 (Hdac6) and ciliary adenylate cyclase 3 (Adcy3) transcript levels. Consistently, microtubule acetylation, endogenous sonic hedgehog (SHH) activation and ciliary ADCY3 were all affected in this group. Application of exogenous GDF15 or pharmacological antagonists of either HDAC6 or ADCY3 similarly normalized ciliary morphology, proliferation and SHH signalling. Notably, *Gdf15* ablation affected Hdac6 expression and cilia length only in the mutant periventricular niche, in concomitance with ciliary localization of GFRAL. In contrast, in the hippocampus, where GFRAL was not expressed in the cilium, progenitors displayed altered Adcy3 expression and SHH signalling, but Hdac6 expression, cilia morphology and ciliary ADCY3 levels remained unchanged. Thus, ciliary signalling underlies the effect of GDF15 on primary cilia elongation and proliferation in apical progenitors.

## Introduction

Growth and differentiation factor 15 (GDF15), a divergent member of the TGF-β superfamily, regulates body weight by controlling food uptake via its receptor GDNF family receptor alpha-like (GFRAL). The latter is sparsely expressed in the healthy adult brain and it is mainly observed in brainstem neurons where signal transduction is induced upon physical interaction with the receptor tyrosine kinase Ret (Emmerson et al, 2017; Yang et al, 2017). However, being ubiquitously expressed, GDF15 has been associated with a range of diseases such as cancer and pathologies of the liver, kidney and cardiovascular system, as well as obesity (Rochette et al, 2020). In the uninjured rat brain, high levels of GDF15 are observed around the lateral ventricle (Böttner et al, 1999; Schober et al, 2001),

suggesting that the factor may affect adult neural stem cells (NSCs) in the ventricular-subventricular zone (V-SVZ). The V-SVZ and the subgranular zone of the dentate gyrus in the hippocampus (HP) represent the main neurogenic niches in the adult mammalian brain. Consistent with an effect of GDF15 on NSC function, GDF15 is expressed in both neurogenic regions (Schober et al, 2001), where it promotes EGF receptor (EGFR) expression and NSC proliferation (Carrillo-Garcia et al, 2014). However, unlike the diffused expression observed in the HP, in the V-SVZ and in its largest embryonic counterpart, i.e., the germinal epithelium of the developing ganglionic eminence (GE), GDF15 and its receptor are particularly expressed on the apical side contacting the lateral ventricle (Schober et al, 2001). Furthermore, unlike in the HP, in the V-SVZ GDF15 prevents proliferation of apical progenitors from late development onwards thereby decreasing the number of ependymal cells and apical NSCs in the adult V-SVZ (Baur et al, 2024). Apical precursors contacting the lateral ventricle extend a primary cilium, which is important for sensing and transducing environmental signals, such as sonic hedgehog (SHH), and for cell cycle progression (Plotnikova et al, 2009; Khatri et al, 2014). Also in the adult V-SVZ, apical NSCs represent the most common population of ciliated progenitors, as the vast majority of NSCs present at the basal side of the neurogenic niche lack the organelle (Baur et al, 2022).

The activity of histone deacetylase 6 (HDAC6), which is responsible for deacetylation of ciliary microtubules, is of primary importance for the dismantling and maintenance of primary cilia during cell cycle progression (Sanchez de Diego et al, 2014; Shi et al, 2021). However, HDAC6 is localized primarily in the cytoplasm and is not strictly a ciliary protein as its activity depends on its subcellular localization and activation (Wang et al, 2016). In contrast, the adenylate cyclase 3 (ADCY3) is essentially localized to primary cilia, and it has been shown to influence primary cilia length and SHH signalling by regulating cAMP levels (Bishop et al, 2007; Ou et al, 2009; Qiu et al, 2016). Notably, alteration in ciliary function and modulation of ADCY3 and GDF15 have all been linked to the regulation of energy homeostasis, food intake, and obesity (DeMars et al, 2023).

Given the expression of GDF15 in the apical niche and its effect on the proliferation of apical progenitors, we have here

[1]Department of Neurobiology, Interdisciplinary Center for Neurosciences (IZN), Heidelberg University, Heidelberg, Germany    [2]Sorbonne University, Paris, France

Correspondence: ciccolini@nbio.uni-heidelberg.de

investigated its effect on primary cilia and the underlying mechanisms. Our data show that GDF15 critically contributes to the regulation of primary cilia in apical progenitors and NSCs in the embryonic GE and in the adult V-SVZ. We also show that this effect correlates with ciliary localization of GFRAL and altered expression/function of HDAC6 and ADCY3.

## Results

### GDF15 ablation changes cilia morphology in the embryonic GE and adult V-SVZ

In a previous study, we found that both GDF15 and its receptor GFRAL are expressed at the apical side of the adult V-SVZ and in the corresponding germinal niche of the GE at embryonic day 18 (E18) (Baur et al, 2024). Because apical cells characteristically extend primary cilia, we investigated if GFRAL is also present in this organelle. Detailed examination of the cilia in E18 WT embryos confirmed the presence of the receptor in cilia identified by immunofluorescence with antibodies against the ADP-ribosylation factor-like 13b (ARL13B) and adenylate cyclase 3 (ADCY3, also known as AC3), both of which are expressed in primary cilia at this age (Monaco et al, 2019). We found that in the apical GE, some ARL13B- and ADCY3-expressing cilia displayed GFRAL immunoreactivity (Fig 1A, white arrows). We also detected a few GFRAL$^+$ cilia expressing only ARL13B (Fig 1A, blue arrows), or both ciliary markers but lacking receptor immunoreactivity (Fig 1A, turquoise arrows). Remarkably, the receptor was also observed in ARL13B$^+$ primary cilia of the adult V-SVZ (Fig S1A). In contrast, a similar analysis on coronal sections of the E18 HP revealed only a rather weak expression of GFRAL and no co-localization with both ciliary markers (Fig 1A, bottom row), which are already expressed in primary cilia in this region at this developmental age (Berbari et al, 2007; Caspary et al, 2007).

Proteins expressed in primary cilia often have a role in ciliary signalling and therefore are also crucial for the morphology and function of the organelle (Pazour et al, 2002; Lechtreck et al, 2008; Escudier et al, 2009; Ou et al, 2009; Larkins et al, 2011; Nakakura et al, 2015; Chen et al, 2016; Bachor et al, 2017; Ehnert et al, 2017; Patnaik et al, 2019; Frasca et al, 2020). To investigate whether GFRAL also affects primary cilia, we used Gdf15-knock-out/LacZ-knock-in (Gdf15$^{-/-}$) E18 embryos to examine whether ablation of the GFRAL ligand affects primary cilia in apical progenitors of the ganglionic eminence (GE). Using immunofluorescence for ARL13B to detect primary cilia (Fig 1B), we found that they were shorter (Fig 1C), thicker (Fig 1D), and more numerous (Fig 1E) in the GE of E18 Gdf15$^{-/-}$ embryos than in WT age-matched control. In the WT apical GE cilia measured on average about 3.67 ± 0.45 $\mu m$ in length, 0.39 ± 0.02 $\mu m$ in thickness, and about 26.7 ± 2.8 cilia per 1,000 $\mu m^2$ were detected. Instead in the Gdf15$^{-/-}$ counterpart, the length and thickness of primary cilia measured 1.84 ± 0.12 $\mu m$ and 0.54 ± 0.01 $\mu m$, respectively. In addition, 38.2 ± 4.1 cilia per 1,000 $\mu m^2$ were counted in the mutant GE. To exclude the possibility that the structural differences were artefacts of immunofluorescence due to changes in protein expression between the two groups, we also examined cilia

morphology using scanning electron microscopy (SEM) (Fig 1F). The cilia length measured in SEM micrographs matched the values obtained from immunofluorescently labelled samples (Fig 1G), whereas the measured overall thickness was decreased (Fig 1H; 0.19 $\mu m$ SEM compared to 0.39 $\mu m$ in fluorescence). However, we still observed a significant difference in thickness between cilia from the WT and Gdf15$^{-/-}$ GE also in this experimental setting. The direct role of GDF15 in cilia morphology was underscored by the fact that exposing whole mount explants of the mutant GE to exogenous human GDF15 for 24 h led to an amelioration of cilia length and thickness without affecting differences in cilia number (Fig 1C–E). To confirm that changes in ciliary length were not a consequence of a different relative localisation between the ARL13B immunopositive portion of the cilia and its ciliary base, we investigated the localization of the basal body with γ-tubulin in the WT and Gdf15$^{-/-}$ apical GE (Fig S1B), which showed no differences between the two groups of tissue. In addition, we could exclude accidental measurement of emerging ependymal cilia because of their distinctive appearance (Fig S1C). Since the difference in cilia number displayed a high variability and was not consistently affected by the treatment, for the rest of the study we focussed only on the analysis of the effect of GDF15 on in cilia morphology. The expression of GDF15 in the GE is up-regulated around E16/E18 and thereafter it remains constant in the postnatal V-SVZ (Carrillo-Garcia et al, 2014). Therefore, we next examined the apical side of the V-SVZ of adult WT and Gdf15$^{-/-}$ mice to investigate whether differences in primary cilia length persist into adulthood (Fig 1I and J). Whereas primary cilia in mutant adult progenitors measuring 3.54 ± 0.18 $\mu m$ were longer than the embryonic counterpart, they were still significantly shorter than primary cilia of adult progenitors in the V-SVZ of WT mice, which measured on average 4.71 ± 0.22 $\mu m$ (Fig 1I, white arrows). Next, we investigated whether GDF15 affects cilia morphology in other brain regions and measured cilia length and number in the HP, hypothalamus, and in the brainstem. We focused on these brain regions because a function for GDF15 and the presence of GFRAL has been previously demonstrated in all of them (Schober et al, 2001; Yang et al, 2017). Using antibodies to either ARL13B or ADCY3 to visualize cilia in the E18 HP (Fig S2A and B) and adult HP (Fig S2C–E), hypothalamus (Fig S2F–H), and brainstem (Fig S2I–K), we found no significant difference in the length (Fig S2B, D, G, and J) or number of cilia (Fig S2E, H, and K) in these brain regions between Gdf15$^{-/-}$ and WT control groups. The number of cilia per cell was not counted in the E18 hippocampus, as the nuclei were too dense to count at this age. Taken together, these data suggest that the effect of GDF15 on primary cilia may require ciliary localized GFRAL.

Upon exposure to its ligand, GFRAL dimerizes and forms a complex with the co-receptor RET, which activates several signalling cascades, including the MAPK-ERK pathway (Mullican et al, 2017) and we have recently shown that GDF15 exposure promotes ERK phosphorylation also in the embryonic GE (Baur et al, 2024). Therefore, to further support the hypothesis that the effect of GDF15 on cilia depends on GFRAL activation, we next examined whether the effect of GDF15 on cilia requires ERK activation using the MEK1/2-inhibitor U0126 to inhibit phosphorylation of ERK. Moreover, to investigate whether it involves changes in gene transcription, we examined whether actinomycin D prevents GDF15 from affecting cilia morphology. Western blot analyses of whole E18 GEs incubated

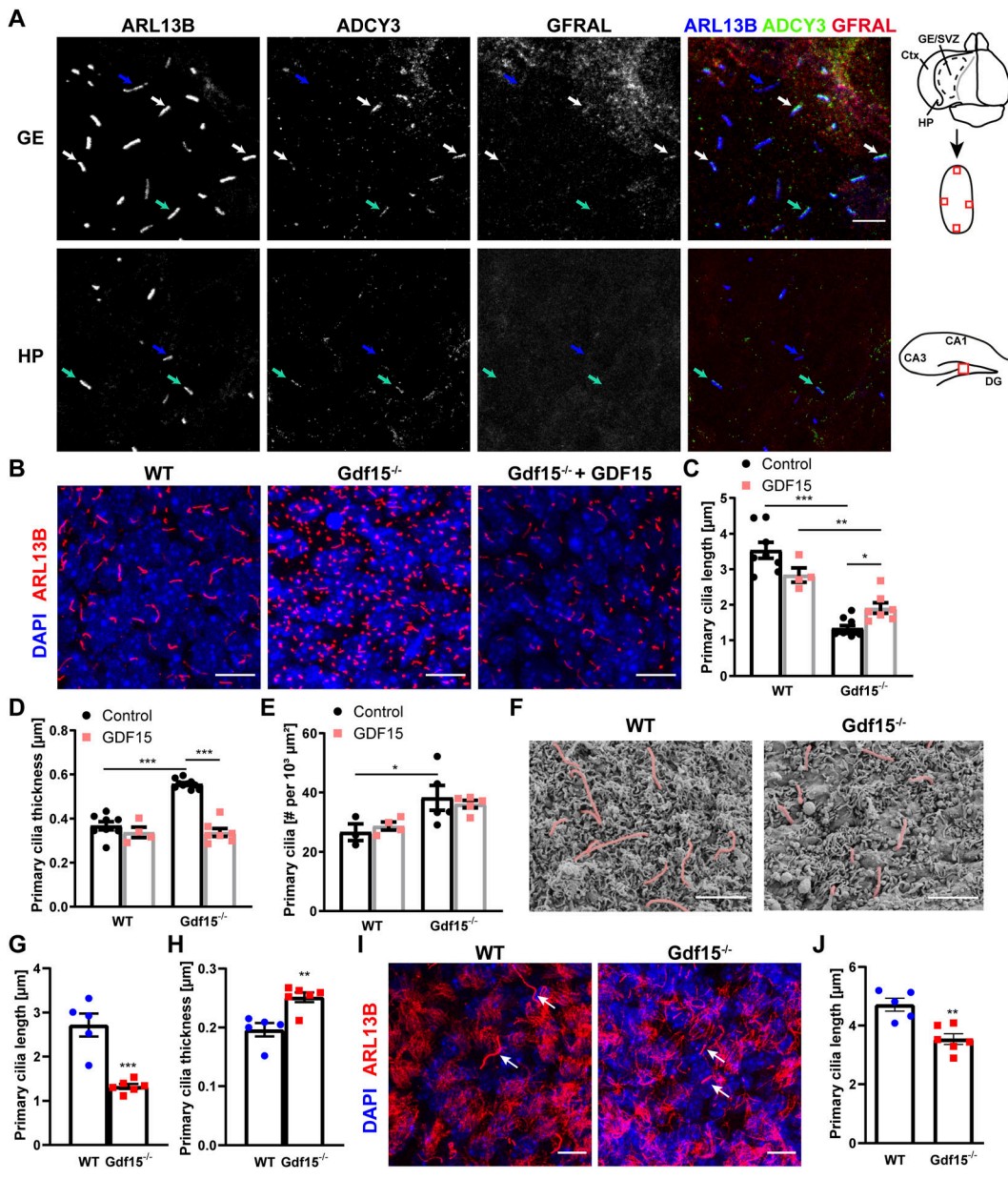

**Figure 1. GDF15 affects cilia morphology and number in the lateral periventricular germinal zone.**
**(A)** Representative confocal micrographs of the apical GE and of the prospective dentate gyrus in HP obtained from whole mount preparations of E18 brain tissue upon triple immunostaining with the ciliary marker ARL13B (blue), ADCY3 (green), and GFRAL (red). White arrows point at cilia expressing both ciliary markers and GFRAL, green and blue arrows to cilia expressing both ciliary markers, or ARL13B only, respectively. Scale bar = 5 $\mu$m. Cartoons show approximate location of the image (red square) in the GE whole mount or hippocampal section. Ctx = cortex, DG = dentate gyrus. **(B, I)** Representative confocal micrographs of whole mount preparations of the GE obtained from E18 embryos (B) and adult V-SVZ (I) of the given genotype and treated with 10 ng/ml exogenous GDF15 as indicated. Tissue preparations were immunostained with antibodies to Arl13b (red) to visualize apical cilia and nuclei were counterstained with DAPI (blue). Scale bars = 10 $\mu$m. **(C, D, E, J)** Quantification of primary cilia length (C, J), thickness (D), and number (E) in immunofluorescent stainings of E18 (C, D, E) and adult (J) tissue of dissected from animals of the given genotype and treated as indicated. **(F)** Representative images of primary cilia (red) at the apical side of the E18 GE of the given genotype upon scanning electron microscopy (ScEM). Scale bars = 2 $\mu$m. **(F, G, H)** Quantitative analysis of cilia length (G) and thickness (H) in SEM samples shown in (F). Bars indicate mean ± SEM. Each data point represents the average of data collected from one individual animal. * indicates significance (*$P$ < 0.05, **$P$ < 0.01, ***$P$ < 0.001) between groups as indicated.

for 24 h in a medium containing GDF15 or EGF, which also activates the MAP-ERK pathway, in the presence or absence of additional U0126, showed as expected that ERK phosphorylation induced by either growth factor was decreased in the presence of U0126 (Fig 2A and B). Parallel tissue samples after the different treatments were

instead fixed and immunolabelled for cilia analysis (Fig 2C). These experiments showed that GDF15 application did not lead to cilia lengthening while MEK1/2 was inhibited, whereas activation of ERK signalling by EGF was not sufficient to rescue cilia morphology. Likewise, blockade of transcription by actinomycin D did not affect

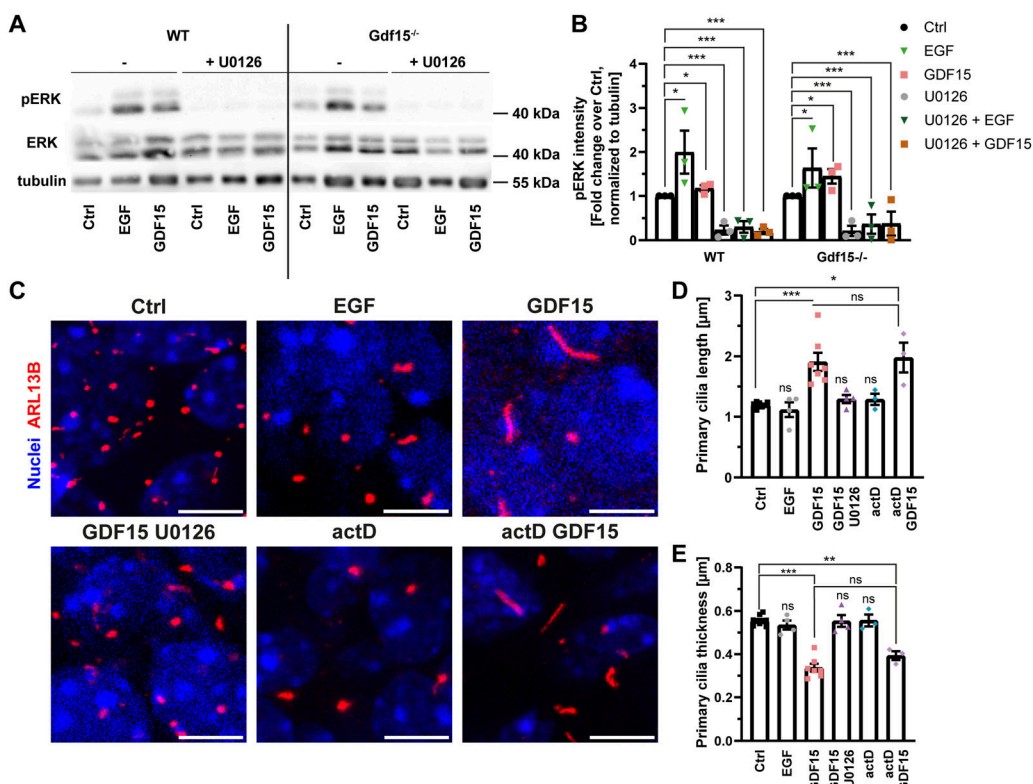

**Figure 2. GDF15 effect on primary cilia requires ERK activation and is transcription independent.**
**(A)** Representative images of Western blot analyses of total (ERK) and phosphorylated (pERK) ERK levels in the whole E18 GE of the indicated genotype cultured in the presence or absence of the given compounds for 24 h. **(B)** Quantification of three independent Western blot experiments showing an increase in pERK upon EGF and GDF15 treatment as well as the reduction of pERK signal by U0126 regardless of genotype and stimulation. pERK levels are shown as fold change over Ctrl, normalized to tubulin as loading control. **(C)** Representative confocal micrographs of primary cilia at the apical side of whole mount preparations of the Gdf15$^{-/-}$ E18 GE upon exposure to the indicated treatments for 24 h. Thereafter, the tissue was processed for immunostaining with ARL13B antibodies (red) and DAPI (blue) nuclear counterstaining. Scale bars = 5 μm. **(C, D, E)** Quantification of the effect of the treatments on length (C) and thickness (D) of primary cilia. Bars indicate mean ± SEM. Each data point represents the average of data collected from one individual animal. * indicates significance (*P < 0.05, **P < 0.01, ***P < 0.001) as indicated. Ns, not significant.

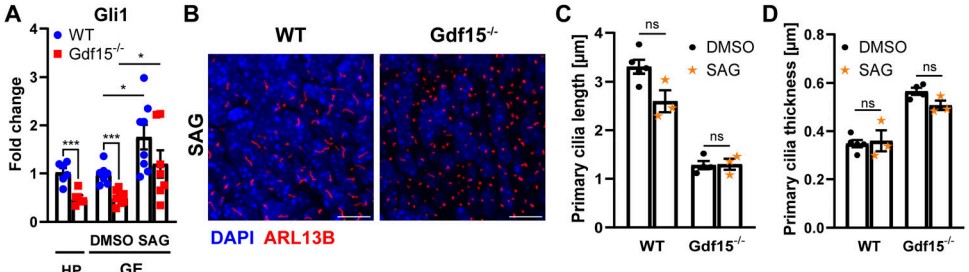

**Figure 3. GDF15 affects endogenous SHH activation.**
**(A)** Fold change in the expression of mRNA Gli1 levels in the whole E18 HP or GE of WT and Gdf15$^{-/-}$ animals, the latter treated with DMSO as control or with smoothened agonist (SAG) as indicated. * indicates significance as indicated (*P < 0.05, ***P < 0.001).
**(B)** Representative confocal micrographs of primary cilia at the apical side of whole mount preparations of the E18 GE of the given genotype treated with SAG for 24 h and processed for immunostaining with ARL13B antibodies (red) to visualize primary cilia and DAPI (blue) nuclear counterstaining. For control, compare to Fig 1B. Scale bars = 10 μm. **(C, D)** Quantification of the effect of SAG on length (C) and thickness (D) of primary cilia. Ns, not significant. Bars indicate mean ± SEM. Each data point represents the average of data collected from one individual animal.

cilia morphology or the ability of GDF15 to rescue the morphological defects of mutant primary cilia (Fig 2D and E). Taken together, these data show that GDF15-GFRAL–dependent activation of MAPK/ERK signalling, but not changes in gene transcription, is necessary for the effect of GDF15 on primary cilia length.

Cilia are known to relay important signals such as SHH signalling. To determine whether the changes in ciliary morphology impact ciliary signalling in the mutant mice, we analysed the mRNA levels of the SHH effector gene Gli1 in the V-SVZ of E18 WT and Gdf15$^{-/-}$ animals using qRT-PCR. We found that in the mutant HP, Gli1 mRNA was reduced by half in comparison to their WT counterparts (Fig 3A). A similar difference was observed in whole mounts of the GE dissected from WT and Gdf15$^{-/-}$ E18 embryos exposed to DMSO (Fig 3A). However, treatment with smoothened agonist (SAG), an SHH pathway activator, led to a similar significant increase in Gli1 transcripts in both groups of progenitors (Fig 3A), albeit it did affect

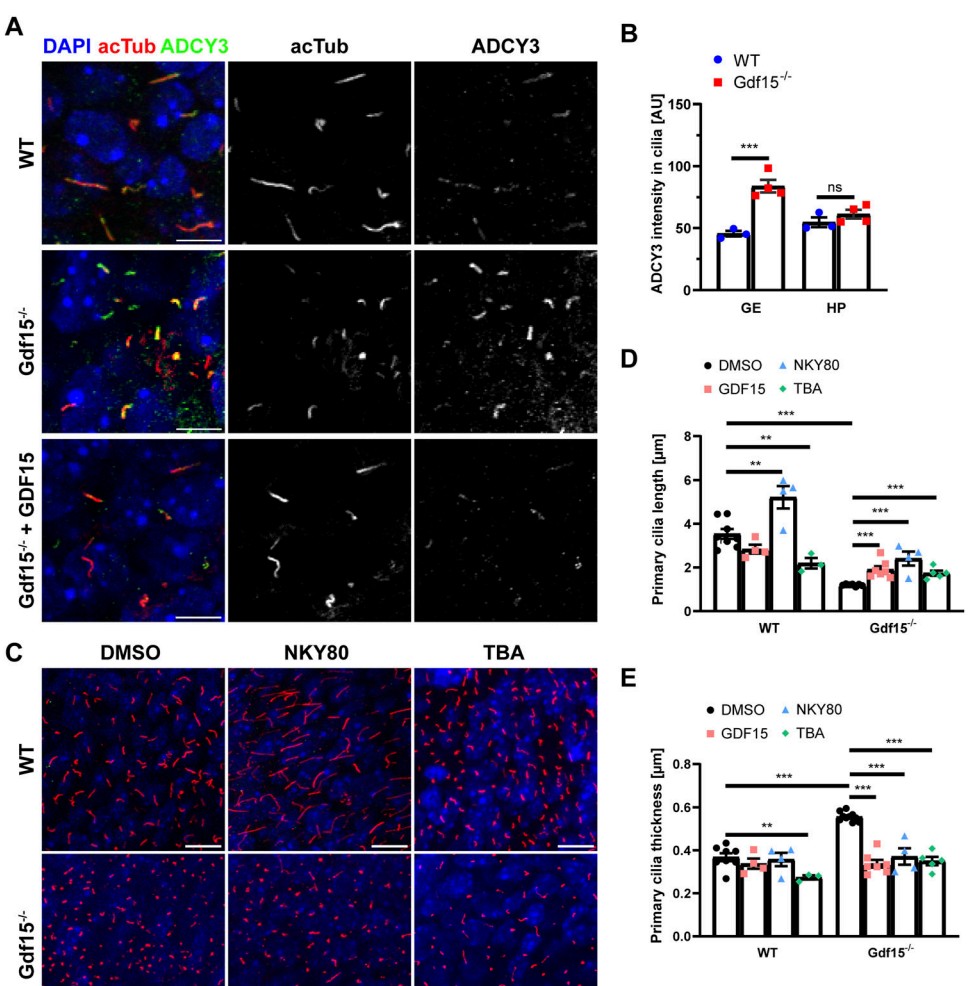

**Figure 4. GDF15 modulates Adcy3 and Hdac6 expression.**
**(A)** Representative confocal micrographs of primary cilia at the apical side of whole mount preparations of the E18 GE of the given genotype upon immunostaining with the indicated antibodies. Note that the expression of ADCY3 but not acTub appeared to be modified according to genotype/treatment. **(B)** Quantification of ADCY3 fluorescence intensity in cilia of whole mount preparations of the GE and coronal sections of the HP of E18 mice. **(C)** Representative confocal micrographs of primary cilia at the apical side of whole mount preparations of the E18 GE of the given genotype treated as indicated for 24 h and processed for immunostaining with ARL13B (red) antibodies to identify primary cilia. DAPI nuclear counterstaining is visualized in blue. Scale bars indicate 10 $\mu$m. **(D, E)** Quantification of the effect of the treatments on cilia length (D) and thickness (E). Bars indicate mean ± SEM. Each data point represents the average of data collected from one individual animal. * indicate significance as indicated (*$P < 0.05$, **$P < 0.01$, ***$P < 0.001$); ns, not significant.

neither the length (Fig 3B and C) nor the thickness of primary cilia (Fig 3D). Taken together, these data indicate that lack of GDF15 affects cilia morphology by a mechanism that is independent of SHH signalling and that mutant progenitors in the GE are still responsive to smoothened relocation.

### ADCY3 and HDAC6 like GDF15 control ciliary length

Endogenous SHH activation, EGFR surface expression, and cell cycle progression are also affected in the apical GE of Gdf15$^{-/-}$ E18 embryos (Baur et al, 2024). To investigate the mechanisms underlying these effects of GDF15, we next analysed whether the expression of ADCY3 and HDAC6 are altered in mutant progenitors. Indeed, besides localizing predominantly to primary cilia of embryonic neural progenitors (Monaco et al, 2019), ADCY3 also promotes the synthesis of cAMP in response to activation of G-protein–coupled receptors, thereby regulating cilia length and SHH signalling (Ou et al, 2009). Moreover, the stability of ciliary microtubules is also affected by $\alpha$-tubulin acetylation mediated by HDAC6, which is important for the shedding of primary cilia before

mitosis but also for the intracellular trafficking of EGFR (Pugacheva et al, 2007; Sanchez de Diego et al, 2014). Quantitative analysis of mRNA levels of Adcy3 and Hdac6 by real-time PCR (qRT-PCR) in the embryonic GE (Fig S3A) and in the HP (Fig S3B) revealed that, compared with the WT counterparts, transcripts for Adcy3 were significantly increased in the absence of GDF15 in the HP and especially in the GE, whereas lack of GDF15 led to increased Hdac6 expression in the GE but not in the HP. Independent of the tissue, treatment with exogenous GDF15 did not affect expression of either transcript in WT samples, whereas it led to decreased mRNAs levels for both genes in the mutant E18 GE and only of Adcy3 transcripts in the HP. In immunofluorescently labelled E18 whole mount preparations of the GE, we also found a visible increase in ADCY3 immunoreactivity in Gdf15$^{-/-}$ animals compared with WT controls (Fig 4A and B). This increase, however, was not observed in primary cilia of the HP, and in the GE was no longer detected upon application of exogenous GDF15 (Fig 4A and B). Notably, in the untreated mutant GE, the increase in ADCY3 expression was observed both in longer and shorter cilia (cilia length above 3 $\mu$m: ADCY3 fluorescence levels WT 41.94 ± 3.48 versus Gdf15$^{-/-}$ 79.98 ± 2.86; cilia length below 3 $\mu$m:

ADCY3 fluorescence levels WT 55.57 ± 4.14 versus Gdf15$^{-/-}$ 88.40 ± 2.76; n ≥ 14; $P$-value < 0.0001; two-way ANOVA). Thus, absence of GDF15 leads to increased Adcy3 mRNA expression and higher expression of ADCY3 protein in primary cilia of the GE, but not in the HP. These conclusions could not be further tested by Western blot analysis due to the poor performance of the available ADCY3 antibodies. Taken together, our data indicate that independent of GFRAL localization, lack of GDF15 leads to an increase in Adcy3 transcripts, whereas increased ADCY3 in primary cilia and change in Hdac6 mRNA levels are observed in mutant tissue displaying ciliary localization of GFRAL. To determine whether the overexpression of ADCY3 and HDAC6 underlies the change in cilia morphology in Gdf15$^{-/-}$ animals, we used the pharmacological agents NKY80 (Fig S3C) and tubastatin A (TBA) to inhibit the function of ADCY3 and HDAC6, respectively. For cilia analysis, E18 whole GE preparations were incubated with or without the agents for 24 h and cilia length and thickness were assessed in immunofluorescent images using ARL13B as a ciliary marker (Fig 4C). Here, we found that both inhibitors led to an increase in length and to a decrease in thickness in the cilia of the apical Gdf15$^{-/-}$ GE, similar to the effect of GDF15 (Fig 4D and E). However, unlike exposure to exogenous GDF15, the treatments also led to a change in cilia morphology in WT progenitors. Whereas NKY80 caused increased ciliary length also in WT progenitors, TBA led unexpectedly to a shortening of primary cilia in this cell group (Fig 4C). Thus, whereas blockade of ADCY3 promotes an increase in cilia length in both WT and mutant progenitors, the effect of HDAC6 activity on cilia length depends on the genotype.

Although exposure to exogenous GDF15 does not require transcription to rescue cilia morphology in mutant progenitors, we have observed that it modulates the expression of Adcy3 and Hdac6 transcripts in the mutant GE, showing a direct involvement of the growth factor in the regulation of the two enzymes. We therefore next investigated whether blockade of either enzyme reproduced a similar effect on transcript expression (Fig S3D and E). This analysis revealed that in mutant tissue, both blockers had a similar effect on transcript levels as exposure to exogenous GDF15. However, both treatments reduced Adcy3 but not Hdac6 levels in WT progenitors, indicating that endogenous GDF15 signalling regulates Hdac6 expression upstream of either enzyme.

## Gdf15$^{-/-}$ and WT apical progenitors display distinct tubulin acetylation patterns

Because HDAC6 function is associated with its intracellular local regulation (Li et al, 2013), the opposite effect of TBA on the length of cilia in WT and mutant progenitors indicates that HDAC6 activity/localization may differ between the two groups of progenitors.

HDAC6 regulates cilia length by deacetylation of ciliary tubulin, thereby shortening the ciliary axoneme, which is also known to affect cell cycle dynamics (Pugacheva et al, 2007; Sanchez de Diego et al, 2014; Ehnert et al, 2017; Shi et al, 2021). Interestingly, inhibition of HDAC6, despite decreasing Adcy3 expression in both WT and mutant progenitors, caused lengthening of cilia only in mutant progenitors but not in the WT counterpart. Because HDAC6 is known to regulate cilia length also by modulating intracellular trafficking and transport, the differential effects observed in the two groups of progenitors may reflect the fact that HDAC6 more effectively

deacetylate ciliary microtubules in mutant progenitors and cytoskeletal microtubules in the WT counterpart. To investigate this hypothesis, we used Western blot to determine the overall levels of acetylated tubulin (acTub) in whole lateral walls of E18 WT and Gdf15$^{-/-}$ animals, with and without incubation with HDAC6-inhibitor TBA, as well as exogenous GDF15 for 24 h (Fig 5A and B). Here, we found that even though HDAC6 is overexpressed in Gdf15$^{-/-}$ animals, WT and mutant tissue displayed similar levels of acTub even after application of exogenous GDF15. In contrast, inhibition of HDAC6 by TBA, as expected, led to a significant increase of acTub in the WT GE, but strikingly not in the Gdf15$^{-/-}$ counterpart where, despite affecting cilia length, blockade of HDAC6 caused only a nonsignificant trend increase in acTub levels (Fig 5A and B). Using immunofluorescence in whole mount preparations, we did not detect significant quantitative differences in HDAC6 expression associated with the genotype or the treatment with TBA, which induced only a non-significant decrease in HDAC6 immunoreactivity in mutant progenitors (Fig 5C and D). On the other hand, analysis of acTub in these preparations confirmed the Western blot analysis, clearly showing that TBA treatment increases acTub at the apical surface of the WT GE but not in the mutant counterpart (Fig 5C and E). Taken together, these data indicate that the change in HDAC6 expression between WT and GDF15 may concern differential activation rather than localization of HDAC6.

## Effect of ciliary length and SHH on proliferation in the mutant apical GE

In a previous publication, we demonstrated that ablation of GDF15 leads to an increased proliferation of embryonic apical and subapical progenitors by increasing the number of cycling cells and by promoting cell cycle progression (Baur et al, 2024). Since the length of primary cilia regulates cell cycle speed (Malicki & Johnson, 2017), it is possible that the changes in proliferation in mutant progenitors are associated with the defect in cilia morphology. Therefore, we next tested whether blockade of HDAC6 or ADCY3, like exposure to exogenous GDF15, beside cilia morphology also affects cell proliferation of mutant progenitors. For this, we used immunofluorescence for the cell cycle marker Ki67 to label cycling cells (Fig 6A) and we also investigated the nuclear morphology of Ki67$^+$ nuclei to score cells undergoing division (mitotic cells; Fig 6A, white arrow). Using this approach, we found that in Gdf15$^{-/-}$ animals, both TBA and NKY80 led to a decrease in the number of total cycling (Fig 6B) and dividing cells (Fig 6C) in the E18 GE, similar to the effect of GDF15. Thus, blockade of HDAC6 or ADCY3 in mutant progenitors recapitulates the effect of GDF15 on cilia morphology and proliferation. In contrast, in the WT GE, the effect of the treatment led only to a reduction of mitosis which was not statistically significant, due to the already lower level of mitosis in this tissue.

Because mutant progenitors display impaired SHH signalling, which may also affect proliferation, we next investigated the effect of SAG on progenitor proliferation. Consistent with previous observations (Komada, 2012) activation of SHH signalling in WT mice increased the number of both cycling and dividing cells (Fig 6D–F). In Gdf15$^{-/-}$ embryos, SAG did not affect the number of total cycling cells (Fig 6E) and instead only led to a significant decrease in dividing cells (Fig 6F), indicating that impaired endogenous SHH

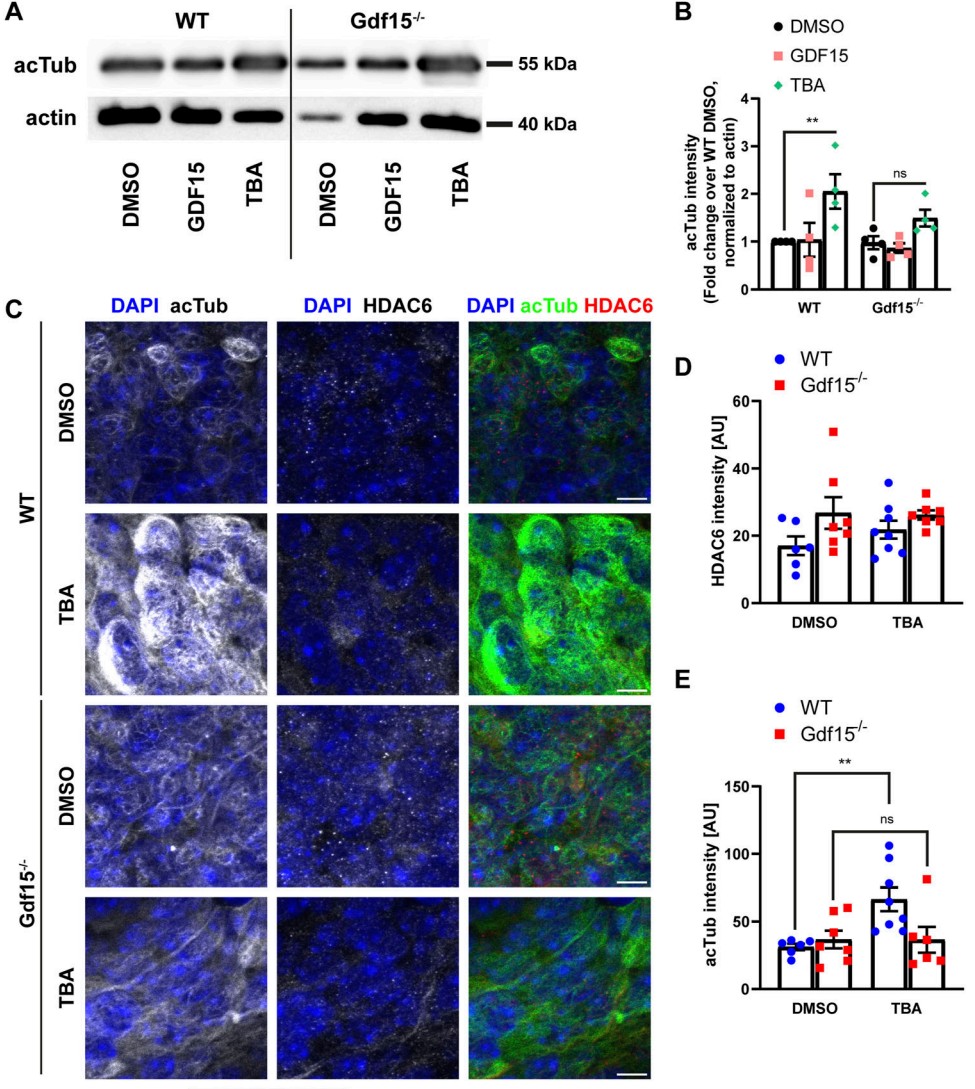

**Figure 5. GDF15 modulates HDAC6 activity. (A)** Western blot for acTub in the whole E18 GE of the indicated genotype incubated in the presence of the given compounds for 24 h. Actin was used as loading control. TBA, tubastatin A. **(A, B)** Quantification of acetylated tubulin (acTub) intensity in (A), normalized to actin. Each data point represents an independent sample from a different animal. **(C)** Representative confocal micrographs of the apical side of whole mount preparations of the E18 GE of the given genotype and upon exposure to the indicated compounds for 24 h. Thereafter the tissue was processed for immunostaining with antibodies to acTub (green), HDAC6 (red), and DAPI nuclear counterstaining (blue). Scale bars = 10 $\mu$m. **(D, E)** Quantification of the effect of the different treatments on HDAC6 (D) and acTub (E) fluorescence levels. Each data point represents the average of data collected from one individual animal. Bars indicate mean ± SEM. * indicates significance (**$P$ < 0.01) as indicated. Ns, not significant.

activation in mutant progenitors is responsible for the increase in the number of cells undergoing mitosis but not for the increase in the pool of cycling progenitors observed in the mutant GE.

We next investigated whether inhibition of HDAC6 and ADCY3, like exposure to exogenous GDF15, also affect *Gli1* expression (Fig 6G). As we have seen that both blockers affect proliferation only in the mutant GE, we performed these experiments only in this group of progenitors. We found that that NKY80 and TBA all led to a significant increase in Gli1 transcripts, an effect especially strong with the latter treatment. To examine the effect on the number of apical progenitors, we determined the number of Prominin-1-expressing (P+) cells in the dissociated GE of E18 Gdf15−/− animals by flow cytometry after 24 h treatment with GDF15, TBA, or SAG, in relation to untreated controls (Figs 6H and S4A–D for FACS gating). Consistent with our previous observations (Baur et al, 2024), treatment with recombinant GDF15 decreased the pool of P+ cells in the mutant GE to about 80% of the cells in untreated samples. In contrast, only treatment with TBA but not with SAG affected P+ cells, indicating that SHH signalling affects cell cycle speed but does not affect the number of apical progenitors in the mutant GE. Taken together, these data show that blockade of HDAC6 and ADCY3 recapitulate the effect of exposure of mutant progenitors to GDF15 both with respect to ciliogenesis proliferation and SHH signalling, whereas up-regulation of SHH signalling rescues only the defect in cell cycle progression of mutant progenitors.

## Discussion

In this study, we report for the first time that the expression of GDF15 in the V-SVZ contributes to ciliary function of apical neural progenitors. Previous studies have highlighted that in embryonic apical radial glia, the inheritance of the mother centriole (Wang et al, 2009) or of ciliary remnants (Paridaen et al, 2013) is associated

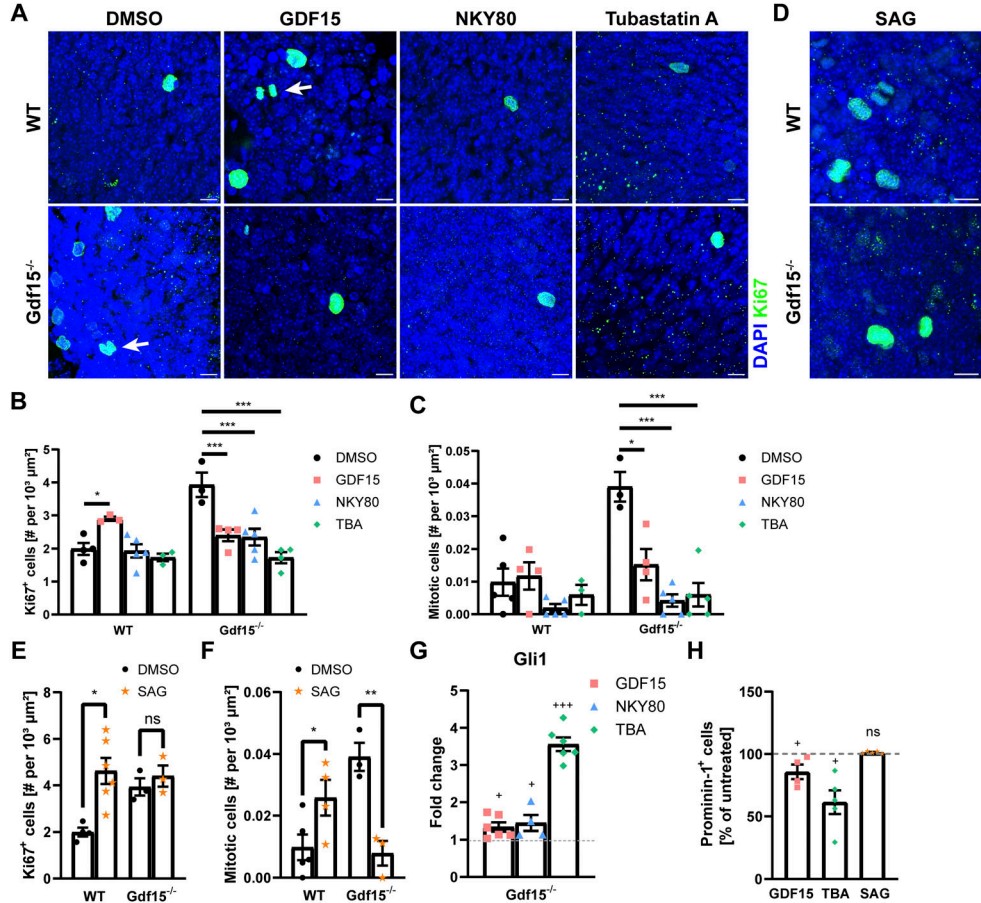

**Figure 6. SHH modulates cell cycle length.**
**(A, D)** Representative confocal micrographs of cycling cells at the apical side of whole mount preparations of the E18 GE of the given genotype. **(A, D)** Tissues were exposed to GDF15, NKY80, or tubastatin A (TBA; (A)), or SAG (D) for 24 h and immunostained with Ki67 antibodies (green) and counterstained with DAPI (blue). **(A)** DMSO was used as control (A). White arrows point to mitotic cells. Scale bars indicate 10 $\mu$m. **(B, C)** Fold change of total cycling (B) and apically dividing (C) cells. **(E, F)** Quantification of total cycling (E) and apically dividing (C) cells after SAG treatment compared with DMSO-treated controls. **(G)** Fold change in the expression of *Gli1* mRNA in the E18 mutant GE after treatment with indicated agents. Values are normalized to the respective untreated sample (=1, dashed line). **(H)** Quantitative analysis of the number of Prominin-1–labelled cells in the Gdf15$^{-/-}$ GE treated as indicated, as determined by flow cytometry. Values represent the percentage of the number of Prominin-1–immunopositive cells compared with DMSO-treated controls (=100%, dashed line). Bars indicate mean ± SEM. Each data point represents the average of data from one individual animal. * indicates significance as indicated (*$P < 0.05$, **$P < 0.01$, ***$P < 0.001$). + indicates significance to DMSO-treated control ($^+P < 0.05$, $^{++}P < 0.01$, $^{+++}P < 0.001$).

with the rapid generation of primary cilia and the maintenance of apical radial glia character at the expense of the generation of more differentiated progenitors. Alteration in the orientation of the plane of division (Falk et al, 2017) or of ciliogenesis (Broix et al, 2018; Hu et al, 2021) can also affect the number of ciliated progenitors; however, these mutations also impact differentiation and the type of progenitors generated upon cell division. In contrast, despite the increased number of primary cilia in the mutant GE, we have previously found no change in differentiation but only an increase in the number of generated cells (Baur et al, 2024), indicating that the increased presence of ciliated cells essentially reflects an expansion of the number of ciliated progenitors. Consistent with this hypothesis, we found that not only apically dividing progenitors but also their progeny, including fast proliferating secondary short progenitors and subapical progenitors, were increased in the mutant GE (Baur et al, 2024). Besides the number of cilia, we report here that mutant cilia display a shorter and thicker appearance. Whereas the importance of the latter to the regulation of the cell cycle is unclear, it is well documented that the length of primary cilia affects cell cycle progression in neural progenitors (Li et al, 2011). A negative correlation between cilia and cell cycle is also

observed in the adult V-SVZ, where the length of primary cilia is associated with increased permanence into quiescence (Khatri et al, 2014). In the adult V-SVZ, cilia are also particularly associated with the apical side of the V-SVZ (Beckervordersandforth et al, 2010; Baur et al, 2022). However, whereas in the germinal niche of the HP primary cilia are essential for the generation and the maintenance of adult NSCs (Breunig et al, 2008; Han et al, 2008; Amador-Arjona et al, 2011), the relevance of primary cilia for the regulation of neurogenesis in the V-SVZ remains elusive. Indeed, ablation of primary cilia in this region appears to reduce proliferation and neurogenesis only in the ventral anterior telencephalon (Tong et al, 2014). A possible explanation for these findings is that primary cilia in the adult V-SVZ are mostly present in a subset of apical NSCs (Khatri et al, 2014), which are slowly cycling cells and do not represent the main contributors of adult neurogenesis (Baur et al, 2022).

Besides cilia length, SHH signalling was also affected in mutant progenitors. Previous observations have shown that ablation of primary cilia in the developing telencephalon from mid-development onwards eliminate Gli1 expression around in the periventricular area, which leads to a decrease in the number of Nkx2.1-expressing cells

(Tong et al, 2014). However, our data show that lack of GDF15 limited endogenous SHH activation also in the HP, where Adcy3 transcript levels, but neither the length of primary cilia nor overexpression of ADCY3 in the cilia was affected. This indicates that increased ciliary expression of ADCY3 is necessary to affect cilia length but not endogenous activation of SHH. Notably, interference with SHH signalling either directly (Komada et al, 2008) or indirectly through modulation of the ciliary function (Wilson et al, 2012) alters cell cycle progression, although the direction of the change may be affected by temporal cues. Our data here support the notion that the effect of SHH on proliferation is context dependent as it increased both the number of cycling and mitotic progenitors in the WT GE, whereas it decreased mitosis in the mutant tissue.

Our data indicate that ADCY3 plays a central role in regulating cilia length in apical progenitors. Not only we found that cilia length was only affected in concomitance with increased expression of ADCY3 in primary cilia but also blockade of ADCY3 led to an increase in the length of primary cilia both in WT and mutant progenitors. In this study, we used as inhibitor NKY80 ($IC_{50}$: 132 $\mu M$) at a comparatively high concentration, i.e., 200 $\mu M$ in whole mounts and 100 $\mu M$ in dissociated cells. Notably, NKY80 is also a potent inhibitor of ADCY5 ($IC_{50}$: 8.3 $\mu M$) and ADCY6 (Brand et al, 2013), both of which localize to primary cilia and have been found to be involved in regulation of ciliary length (Besschetnova et al, 2010; Arora et al, 2020). Therefore, it is possible that some of the effects seen upon application of NKY80 may be caused by its inhibition of ADCY5/6, both of which were not further analysed in this study due to a lack of reliable commercial antibodies and their relatively unknown role in primary cilia in the brain. However, their role in this context may be significant and warrants further exploration.

We found that the effect of GDF15 on ciliary function is limited to the V-SVZ, thereby providing a possible explanation for the differences of the growth factor in the regulation of proliferation in the two neurogenic regions (Carrillo-Garcia et al, 2014). Indeed, in these previous studies, we have shown that late in development, the expression of GDF15 increases in the dentate gyrus and in the GE and that in both brain regions, GDF15 ablation leads to a decrease in EGFR expression at the cell membrane. In both neurogenic regions, EGF is the main mitotic factor for NSCs, making its expression a reliable marker for the identification of these cell populations in both neurogenic regions. However, how EGFR signalling contributes to proliferation is still unclear because the effect of EGFR activation on proliferation varies according to age, and multiple intracellular pathways are activated contributing to different functional outputs including survival and differentiation (Cochard et al, 2021). Notably, it has been shown that HDAC6 affects EGFR trafficking and degradation (Gao et al, 2010) and that absence of the deacetylase leads to faster degradation of the receptor. Because our data indicate a difference in HDAC6 function in apical WT and mutant progenitors, it is possible that this may also be responsible, at least in the GE, for the difference in EGFR expression between the two groups of cells (Sanchez de Diego et al, 2014). Notably, Adcy3 expression was altered also in the mutant HP, which displayed normal cilia. This indicates that local activation of ADCY3 may be necessary to induce the alteration in ciliogenesis. Interestingly, we have found that GFRAL is localized to primary cilia only in the GE/V-SVZ, but not in

the HP, suggesting the exciting possibility that the receptor may affect the activation of ADCY3 within the ciliary space. However, local clues reflecting intrinsic and extrinsic differences may also play a role. For example, whereas NSCs in both neurogenic region express Prominin-1 (Carrillo-Garcia et al, 2010; Walker et al, 2013), the latter like GFRAL is present in primary cilia of NSCs only in the V-SVZ but not of the HP (Walker et al, 2013).

In light of the fact that GDF15 is expressed apically in the lateral ventricle from late development onwards, our data indicate that the growth factor may contribute to the decline in the number of ciliated apical cells and increase in cilia length observed in this region during aging. However, the precise mechanism of action is still unclear and should be further explored.

# Materials and Methods

## Animals

All animal experiments were approved by the Regierungspräsidium Karlsruhe and the local authorities at Heidelberg University. The Gdf15$^{-/-}$ line was previously described (Strelau et al, 2009) and bred with homozygous animals with regular blood refreshing. Animals were mated for 48 h and then separated or separated 24 h after the appearance of a vaginal plug. The separation day was considered as embryonic day 1 (E1).

## Whole mount dissection and incubation

Whole lateral walls were dissected and fixed as described before (Mirzadeh et al, 2008; Monaco et al, 2019). In short, the mice were killed by $CO_2$ inhalation and subsequent cervical dislocation (adult) or by decapitation (E18), and the brain was dissected in a buffered sucrose solution. The lateral wall was thinly removed and either directly fixed in ice-cold 3% PFA, 4% sucrose in PBS, or incubated at 37°C overnight in 1 ml Euromed-N (#ECM0883L; Euroclone) containing 2% B27 (#17504044; Invitrogen) and, when indicated, recombinant human GDF15 (10 ng/ml; #Q99988; R&D Systems), human recombinant EGF (20 ng/ml; #AF-100-15; Peprotech), PD158780 (20 $\mu M$; #513035; Calbiochem/Merck), AMD3100 octahydrochloride hydrate (6 $\mu M$; #A5602; Sigma-Aldrich), TBA (10 $\mu M$; #S8049; Selleckchem), NKY80 (200 $\mu M$; #116850; Sigma-Aldrich), U0126 (10 $\mu M$; #BML-EI282-0001; Enzo Life Sciences), or SAG (200 nM; #11914; Cayman Chemicals). For immunofluorescence, the incubated lateral walls were fixed the next day, and all dissections were left in the 3% PFA, 4% sucrose in PBS solution at 4°C overnight, and then kept in PBS containing 0.01% azide until immunostaining.

## Brain sections

For brain sections, mice were killed as for the whole mount preparation; subsequently, the entire brain was removed and fixed in 4% PFA in PBS for 72 h at 4°C with slight agitation. For cryoprotection, the brains were then submerged in 30% sucrose in PBS solution for 48 h. The cryoprotected brains were then frozen at −20°C and sliced into 20-$\mu$m-thick sections using a Leica CM1950

cryostat. Coronal or sagittal slices were rinsed in PBS and stored in PBS containing 0.01% azide until immunostaining.

### Immunostaining

Immunostaining of brain sections and whole mounts was performed as previously described (Luque-Molina et al, 2019; Monaco et al, 2019). For antibodies used, please refer to Table S1.

### SEM

SEM was performed as described before (Monaco et al, 2019).

### Real-time qRT-PCR

For real-time quantitative PCR (qRT-PCR), RNA was extracted using a Quick-RNA Microprep Kit (#R1054; Zymo Research). For this, the dissected tissue (GE/V-SVZ or HP from one hemisphere, or brain stem and hypothalamus) was lysed in 300 µl lysis buffer; the RNA was extracted according to the manufacturer's instructions, including a DNase digestion step. The RNA was eluted from the column in 20 µl RNase-free water and then retrotranscribed into cDNA using M-MLV reverse transcriptase (#M1701; Promega) and Oligo(dT)15 primers (#C1101; Promega) according to the manufacturer's instructions in a ProFlex PCR machine (Applied Biosystems).

1 µl cDNA was used for qRT-PCR in a volume of 20 µl using TaqMan GeneExpression Master Mix (#4369016; Thermo Fisher Scientific) according to the manufacturer's instructions using a StepOne Plus Real-Time PCR System (Thermo Fisher Scientific). Probes used are listed in Table S2.

For normalization, beta-actin was used as a housekeeping gene; fold change was calculated as $2^{-\Delta\Delta C_T}$, where $\Delta C_T$ is the difference between cycle threshold ($C_T$) value of the gene of interest to its respective beta-actin $C_T$ value and $\Delta\Delta C_T$ is the difference between the $\Delta C_T$ value of a control sample (e.g., WT untreated) and that of the sample of interest. All samples were run as duplicates.

### Western blot

After incubation with inhibitors for 24 h (see "Whole mount dissection and incubation"), whole GEs were dissociated by pipetting in 600 µl RIPA buffer containing 1x protease inhibitor (complete EDTA-free, #11873580001; Roche). 15 µl of the sample was mixed with 5 µl 4x Laemmli sample buffer (62.5 mM Tris–HCl, pH 6.8, 10% glycerol, 1% SDS, 0.005% bromophenol blue), heated to 95°C for 5 min, and then Western blot was performed as previously described (Baur et al, 2024). Briefly, the samples were run on a 7.5% SDS acrylamide gel at constant current and subsequently transferred to a nitrocellulose membrane using a wet transfer system. The membrane was blocked in 5% milk in Tris-buffered saline containing 1% Tween-20 (TBS-T) and primary antibodies were applied in TBS-T overnight. After washing, HRP-conjugated secondary antibodies were incubated in 5% milk in TBS-T for 1 h at room temperature.

For a list of primary and secondary antibodies used, see Table S3.

For densitometry analysis, band luminescence level was measured using ImageJ; bands were first normalized to their respective loading control (actin or α-tubulin) and then to the untreated control or the WT sample as indicated in the figures.

### cAMP assay

For cAMP analysis, the dissected E18 GE of WT and Gdf15$^{-/-}$ animals was dissociated by mechanical trituration and cells were plated at 20,000 cells per well (triplicates for each animal) in a 96-well plate coated with poly-D-lysine in 100 µl Euromed-N containing 2% B27. The cells were left to attach overnight in an incubator at 37°C, 5% $CO_2$. The next day, the medium was removed and replaced by 20 µl induction buffer (Krebs-Ringer bicarbonate buffer containing 500 µM IBMX [#2845; Tocris]) and 100 µM Ro 20-1724 (#B8279; Sigma-Aldrich) with either 10 µM Forskolin (#F3917; Sigma-Aldrich), 100 µM NKY80 (#116850; Sigma-Aldrich), or DMSO as control, and the cells were further incubated for 30 min at 37°C, 5% $CO_2$. Afterwards, cAMP levels were measured using a cAMP-Glo Assay (#V1501; Promega) according to the manufacturer's instructions (protocol for adherent cells, technical bulletin TB357, revised 6/17). Bioluminescence was measured using a CLARIOstar Plus plate reader (BMG Labtech). The cAMP concentration was determined based on a standard curve as described in the manufacturer's protocol. Fig S3C shows data from two independent experiments, each data point indicating the average value of triplicates from a single animal.

### Flow cytometry

Flow cytometry was performed as previously described (Luque-Molina et al, 2019; Baur et al, 2022). Cells were labelled with rat-anti-Prominin-1 antibody (Brilliant Violet 421-conjugated; #141213, Lot #B255870, 1:1,000; BioLegend).

### Imaging and analysis

The immunolabelled tissues were imaged using a Leica SP8 confocal microscope as described before (Baur et al, 2024) and analysed with Fiji/ImageJ (Schindelin et al, 2012); calculations were done using Microsoft Excel.

#### *Primary cilia*
At least 300 apical cilia were measured for each animal by analysing at least 100 cilia in at least three different regions of the whole mount (see also Fig 1A, cartoon). For hippocampus measurements, four images of identical regions in the DG were taken for each animal, and all visible cilia analysed. Numbers displayed in graphs represent the average length/thickness for all analysed cilia of one animal (from one whole mount, 3–5 images, 300 to over 1,000 cilia). For the cilia number, the sum of all counted cilia was divided by the sum of the total area in which the cilia were counted for each animal. To measure the cilia length, apical cilia were marked on partial projections of confocal z-stacks using the ImageJ straight line or segmented line selection tool; the line was drawn from one to other end along the centre of the ciliary axoneme. For thickness measurements, lines were drawn perpendicular to the longitudinal axes of the cilia at the point where individual cilia showed the

largest thickness. To assess the number of cilia in the V-SVZ, we counted only those cilia above the apical surface, i.e., an area without visible DAPI staining, inside one representative 50 × 50 $\mu m$ square per image and normalized on this area. In the DG, hypothalamus (HTh), and brain stem (BSt), cilia in all layers of a z-stack were counted and normalized on the number of nuclei. Note that independently of the method used, i.e., confocal microscopy or SEM (see below), the measured length and thickness, although useful to compare changes in morphology between the different cilia groups, represent approximation of the real values because of the limitations inherent in both methods. All cilia and the nuclei in the DG were counted manually using the "Cell Counter" ImageJ plugin, whereas nuclei in the HTh and BSt were counted using an ImageJ-macro using the "Analyse Particles" function as well as the "Adjustable Watershed" plugin. Analysis of ADCY3 expression in cilia was carried out by highlighting the cilia with the segmented line tool and measuring the fluorescence level using the measure function.

### Cell proliferation

Cell proliferation was assessed in at least four different regions of the whole mount (see also Fig 1A, cartoon). In each image, every visible Ki67[+] cell was counted. Apical and subapical dividing cells were scored as described before (Baur et al, 2024).

### Statistical analysis and graphs

Statistical analysis and graphing were performed with GraphPad Prism 8. Statistical significance was determined by a two-tailed $t$ test for two and by two-way ANOVA with Dunnett's or Sidak's multiple comparisons test for more than two groups with the same or different N numbers, respectively. Significance was reached at *$P$ < 0.05, **$P$ < 0.01, ***$P$ < 0.001. All bars represent the mean value ± SEM. The data points indicate values of individual animals, i.e., the N number is represented by the respective number of data points in each bar.

## Data Availability

No large datasets were generated in this study.

## Supplementary Information

## Acknowledgements

K Baur was supported by the Interdisciplinary Center for Neuroscience (IZN) and the Landesgraduiertenförderung (LGF) of the Heidelberg University Graduate Academy. The authors would like to acknowledge the Electron Microscopy Core Facility (EMCF) at Heidelberg University.

## Author Contributions

K Baur: data curation, formal analysis, methodology, and writing—original draft, review, and editing.
Ş Şan: data curation, investigation, and methodology.
G Hölzl-Wenig: formal analysis and methodology.
C Mandl: formal analysis and methodology.
A Hellwig: data curation and methodology.
F Ciccolini: conceptualization, data curation, formal analysis, supervision, funding acquisition, investigation, project administration, and writing—original draft, review, and editing.

## Conflict of Interest Statement

The authors declare that they have no conflict of interest.

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
