## [Reviewer comments · Life Science Alliance]

Life Science Alliance

GDF15 controls primary cilia morphology and function thereby affecting progenitor proliferation

Katja Baur, Şeydanur Şan, Gabriele Hölzl-Wenig, Claudia Mandl, Andrea Hellwig, and Francesca Ciccolini

DOI: <https://doi.org/10.26508/lsa.202302384>

Corresponding author(s): *Francesca Ciccolini, Heidelberg University*

Review Timeline:

Submission Date:	2023-09-19
Editorial Decision:	2023-11-07
Revision Received:	2024-04-17
Editorial Decision:	2024-04-19
Revision Received:	2024-04-25
Accepted:	2024-04-26

Transaction Report:

November 7, 2023

Re: Life Science Alliance manuscript #LSA-2023-02384-T

Dr. Francesca Ciccolini
Heidelberg University
Department of Neurobiology, Interdisciplinary Center for Neurosciences (IZN), University of Heidelberg
INF366
Heidelberg 69120
Germany

Dear Dr. Ciccolini,

Thank you for submitting your manuscript entitled "GDF15 controls primary cilia in the V-SVZ thereby affecting progenitor proliferation." to Life Science Alliance. The manuscript was assessed by expert reviewers, whose comments are appended to this letter. We invite you to submit a revised manuscript addressing the Reviewer comments.

Thank you for this interesting contribution to Life Science Alliance. We are looking forward to receiving your revised manuscript.

Sincerely,

B. MANUSCRIPT ORGANIZATION AND FORMATTING:

Reviewer #1 (Comments to the Authors (Required)):

This study aims to examine the role of Growth/differentiation factor 15 in the regulation of primary cilia. Using V-SVZ tissues from wild type and GDF15^{-/-} mice, the authors undertake a series of immunostainings to evaluate cilia number and morphology, specifically cilia length and "thickness", and demonstrate cilia number is increased, and morphologically are shorter and thicker with ablation of Gdf15. The authors demonstrate rescue of the cilia phenotype with exogenous Gdf15, and treat the tissue with a series of inhibitors or agonists to identify the molecular mechanisms underpinning the aberrant cilia phenotype.

Major comments

Using inhibitors and agonists such as EGF and Shh, the authors make claims about how the various signaling pathways or signaling components regulate the cilia length or thickness in the Gdf^{-/-} mice. However, in this reviewer's opinion, the images presented do not reflect the quantitative data presented (specific examples detailed below). Furthermore, the spread of the data points in several of the graphs is quite large, raising concerns about the conclusions being drawn from the data. In addition, there are "" above some bars on several of the graphs, but it is unclear what condition it is being compared to, and therefore, what claim of statistical significance the authors are making.

Were outliers removed from the data sets? If not, why not?

According to the methods, each data point on the bars in the graphs represents one "n", which is from one animal. However, this reviewer would like a description of how many cells/cilia/fields of view were scored in each animal. Furthermore, in the figure legends the authors write "each data point represents summarized data from one animal" - what is meant by "summarized"? The manuscript was difficult to read and follow at times, made harder by the use of non-scientific language, poor grammar and long sentences. Moreover, the authors make conclusions based on the data from this manuscript coupled with data from a recent previous article of theirs (Baur et al., 2023), however, their recent previous work is not peer reviewed or published, it is in BioRxiv. In addition, this study does not have a series of cumulative evidence supporting their interpretations/conclusions as is typical in research articles. Consequently, the conclusions drawn are not well supported.

Figure 1

1. Can authors provide explanation for the differential localization of the ciliary proteins in Figure 1? Perhaps counting the GFRAL-positive cilia relative to the other ciliary proteins will reveal a
2. Figure 1E is unclear. The authors claim cilia number is at WT levels in the Gdf^{-/-}+GDF15, however, the graph (1E) does not appear to reflect this. There is also an internal consistency, as the authors report cilia number is 23.44, but the graph (1E) indicates it is closer to 40. In addition, there is a statistical "" above the Gdf^{-/-} control bar (bar 3), but it is not clear what that is being compared to.
3. Figure 1F is not labelled in the Figures.

Figure 2

1. How many times were the Western blots performed? How many tissue samples were analysed ie. What is the n? Densitometry is necessary to quantify the reported differences in pERK. At a minimum, tissues samples from 3 animals should be analysed.
2. The wild-type and Gdf15^{-/-} control
3. It is unclear how the authors can confidently identify the cilia axoneme "length" versus "thickness" in the samples with stunted cilia ie. actD, GDF15 U0126 etc. Labelling the basal body to provide cilia orientation would make the data more compelling.
4. The quantification in 2C/D is inconsistent with the corresponding images presented in 1B and 2B. For example, the EGF and actD images in 2B shows elongated cilia and stunted cilia, respectively, however, the spread of the data points/quantification do not reflect this. Furthermore, why wasn't cilia number scored for the 2B images, particularly as some of the treatments appear to show a reduction in cilia number compared to the Gdf^{-/-} in 1B? In addition, as EGF induces pERK, perhaps treatment of GDF^{-/-} with EGF would rescue or partially rescue the ^{-/-} phenotype.
5. The authors report their previous studies have identified reduced membrane EGFR in Gdf15^{-/-}, therefore, is EGF an appropriate agonist if the receptor levels are reduced in the ^{-/-}? Moreover, the previous studies the authors reference are seemingly not published, the work is reported in BIORXIV

Figure 3

1. The spread of data in the SAG-treated samples in 3A raises concerns about how meaningful and significant the results are. Rather than using whole tissues, can the cell populations be isolated/cultured/treated in vitro?
2. Although the SAG-induced expression of Gli1 is reportedly reduced in the Gdf^{-/-} compared to WT, the relative increase in

Gli1 in the SAG-treated GDF^{-/-} compared to the DMSO treated Gdf^{-/-}, appears similar to the WT ^{-/+} SAG treatment. Therefore, the responsiveness of GDF^{-/-} cells to SAG, may be similar to the WT.

3. What was the rationale behind investigating whether SAG treatment (Shh signaling) impacted cilia length or thickness in the ^{-/-}? Did the WT show any changes in cilia dimensions with SAG treatment? If Gdf15 acts downstream of SAG (Smoothed) to modulate cilia dimensions, it would not be evident in the assay the authors have done. The authors have over interpreted this data.

Figure 4

1. It is unclear which image shows the increased ADCY3 immunoreactivity - there is no figure citation. If the authors claim an increase in immunoreactivity, it should be quantified.
2. The authors claim the increase in ADCY3 immunoreactivity was "no longer detected upon application of exogenous GDF15, indicating that the increased in ADCY3 immunoreactivity in ADCY3 in the GE may reflect the shortening of primary cilia." Can authors explain this claim? What is meant by "reflect". Moreover, the apparent ADCY3 immunoreactivity results correlate with the claimed mRNA transcript levels of ADCY3 before and after GDF15 treatment. Therefore isn't that the most likely explanation?
3. The rescue of the cilia phenotype using NKY80 or Tubastatin A is not convincing based on the images presented in 4D. Moreover, the images and the corresponding graphs are inconsistent with one another.
4. Please clarify the conclusion on line 330/331, what "signaling molecules" are being referred to?

Figure 5B

1. Was the densitometry measured relative to a loading control?
2. The authors seem to propose HDAC6 may more active at cilia in the Gdf^{-/-}, however, all of the analysis is at the population level ie. Western blot and immunofluorescence of whole tissue. Moreover, the HDAC6 tissue staining was not co-localised with a cilia marker. It is unclear how the author expect to identify local changes at cilia using these approaches.

Figure 6

1. In the results section relating to Figure 6, the figure citations direct the reader to Figure 4 in lines 374-376
2. The authors claim to examine a "causal" relationship between cilia and proliferation by treating tissues with inhibitors and seeing if it leads to a decrease in "cell cycle speed" via Ki67 staining. (line 369-374). This approach does not evaluate a "casual" relationship - it is correlative. To demonstrate if it is causal, the phenotype needs to be rescued by modulating the candidate signaling pathway. Moreover, cell cycle speed can not be accurately evaluated via ki67 staining. Were the cells/tissues synchronized before commencing the treatments? Can the cells be isolated, cultured in vitro and the assay performed?
3. Why are there so few Ki67-positive cells in the tissue? It appears there are far more ciliated cells than proliferating cells.

Referee Cross-Comments' to your review report

I agree with the other reviewers comments.

Reviewer #2 (Comments to the Authors (Required)):

Baur et al. demonstrated that Growth/differentiation factor 15 (GDF15) regulates the morphology of primary cilia in apical progenitors in the subventricular zone (SVZ). The authors also indicated that the GDF15-HDAC6 axis plays a critical role in the regulation of primary cilia morphology, affects sonic hedgehog (Shh) signaling and the proliferation of apical progenitors. This paper shows an interesting aspect of neural stem cell regulation by primary cilia. However, I find the manuscript somewhat immature, and it was difficult for me to read it. I think the authors should rework the text and reorganize the figures.

Major points.

1. It is very difficult for me to read the manuscript. Each sentence is too long and contains many messages. Please rewrite the results section and make it easy to read.
2. NKY80 is used as 200 uM in the study. The IC50 of NKY80 for adenylyl cyclase (AC3) is about 132 uM. I feel very uncomfortable with this compound because of the high dosage. Moreover, the IC50 of NKY80 for AC5 is 8.3 uM, and some researchers suggest that AC5 and AC6 are also localized in the cilia. Please confirm that the effect of NKY80 is mediated by AC3, but not any other ACs.
3. The order of the figures in the text is difficult to follow. For example, I can't find Figure 1B in the text. Please check the orders in which the figures are listed.
4. It is difficult to understand the high mag images. For example, Fig 1A is an image of the germinal eminence and hippocampus at E18. However, I have no idea where the images were taken from. Please clarify the location by showing low mag images or cartoon.
5. I am very surprised that not all Arl13B positive cilia have ADCY3 in Figure 1A. Did the authors confirm this observation with another ADCY3 antibody? Have the authors used Rabbit Polyclonal Antibody to Adenylate Cyclase III (EnCor Biotechnology Inc, Cat# RPCA-ACIII)?
6. It is difficult to interpret the images of primary cilia in Figure 1B. The number of primary cilia in GDF15 KO mice is about 5 times higher than the control images. However, Figure 1E shows that the number is less than 2-fold increase. Please clarify this discrepancy. In Figure 1E, the addition of GDF15 did not change the number of GDF15 KO cilia. However, the number of cilia from GDF15 KO+GDF15 is less than GDF15 KO cilia in Figure 1B.
7. I am concerned about the picture of thick cilia in GDF15 KO mice in Figure 1B. I understand that the authors showed SEM

- images to clarify the thickness. Please include the SEM images in Figure 1. Please also include the high mag image of cilia. Please clearly show the picture of cilia of GDF15 KO mice is not over saturated.
8. Since the authors showed 4 graphs in Figure 1C-E, please show each picture in Figure 1B.
 9. How do you know that the cilia you counted are bona fide primary cilia in Figure 1F and 1G? Please clearly show that these are primary cilia, and not cilia from developing ependymal cells. I think the authors can co-stain with GFAP and beta-catenin to show ependymal cells and type B cells. I think the authors can co-stain with CD133, as shown in the previous paper (Khatri et al., Scientific reports 2014)
 10. Please include pictures of control and GDF15 in Figure 2B, as shown in Figure 2C and 2D.
 11. Figure 3A is difficult to interpret. I think the "+" indicates the difference between DMSO treatment in GE and SAG treatment in GE. Please show a bar to clearly indicate which graphs you have analyzed for statistics. Please also include "ns (not significant)" for important bar graph comparisons, such as SAG treatment in WT GE and SAG treatment in GDF15 KO GE. Please do the same for Figures 4A, 4E, 4F, 6B and 6C
 12. Please include bar graphs of WT DMSO and GDF15 KO DMSO in Figure 3C and 3D.
 13. It is difficult to interpret Figure 4C. The WT picture clearly shows ADCY3 localization in most of the cilia in contrast to Figure 1A. I agree that GDF15 KO cilia have increased levels of ADCY3. I think the authors need to quantify the intensity of ADCY3 in Figure 4C. The picture of GDF15 also makes me wonder if the thickness is really different as compared to the control, because it looks like some cilia are oversaturated. As you know, oversaturation affects the thickness of the cilia.
 14. In Figure 5C, it is difficult to interpret the HDAC6 staining. Are these signals real? The authors can use antigen retrieval protocol for immunohistochemistry to obtain better signals.
 15. Is there a way to show that both NKY80 and Tubastatin A are really working in Figure 4 and 5?
 16. In Figure 6a, I am surprised at the number of Ki67 positive cells you show in the pictures. I thought there should be more proliferating cells in that area. How did you stain the Ki67 antibody? I thought the abcam rabbit Ki67 antibody works best for room temperature overnight condition. The authors can use other Ki67 antibodies, pH3 or PCNA antibodies to confirm the results.
 17. In Figure 6A, please include pictures of GDF15 treatment, as the bar graphs are shown in Figure 6B and 6C.
 18. In Figure 6B and C, please include DMSO treatment graphs as bar graphs.
 19. In Figure 6C, the authors may use pH3 staining to show the mitotic cells,
 20. In Figure 6D, please include DMSO treated pictures.
 21. In Figures 6E, 6F, 6G and 6H, please include DMSO-treated bar graphs.
 22. In Figure 6H, please clarify the meaning of CD133+. Is this due to decreased number of apical progenitors, or apical progenitors without CD133 in the cilia? Some receptors may exit cilia after signal activation. Please include raw data of flow cytometry data of CD133 in the Supp figures. I would like to see all gating, such as FSC, SSC, DAPI and CD133. Please include isotype control and make a tight gating to clearly show that the CD133 signal is real.

Minor points.

1. Please include catalogue numbers in the method section.
2. Please include the molecular size in the figure of western blot pictures.

Reviewer #1 (Comments to the Authors (Required)):

This study aims to examine the role of Growth/differentiation factor 15 in the regulation of primary cilia. Using V-SVZ tissues from wild type and GDF15^{-/-} mice, the authors undertake a series of immunostainings to evaluate cilia number and morphology, specifically cilia length and "thickness", and demonstrate cilia number is increased, and morphologically are shorter and thicker with ablation of Gdf15. The authors demonstrate rescue of the cilia phenotype with exogenous Gdf15, and treat the tissue with a series of inhibitors or agonists to identify the molecular mechanisms underpinning the aberrant cilia phenotype.

Major comments

Using inhibitors and agonists such as EGF and Shh, the authors make claims about how the various signaling pathways or signaling components regulate the cilia length or thickness in the Gdf^{-/-} mice. However, in this reviewer's opinion, the images presented do not reflect the quantitative data presented (specific examples detailed below). Furthermore, the spread of the data points in several of the graphs is quite large, raising concerns about the conclusions being drawn from the data. In addition, there are "" above some bars on several of the graphs, but it is unclear what condition it is being compared to, and therefore, what claim of statistical significance the authors are making. Were outliers removed from the data sets? If not, why not?*

As the reviewer mentions we have used immunostaining and extensive quantification to analyze the effect of GDF15 and other signalling molecules on cilia morphology, appearance and cilia signalling. Each data set was obtained through the analysis of at least 300 individual cilia per animal and data were collected from various region of interest through a standardized procedure to take into account of regional variability. As the reviewer mentions, we were not always able to match this extensive characterization with representative images as they reflect only regional snapshots: We would like to apologize for these shortcomings and we have now replaced several panels in the figures, in an effort to improve their representativeness. Because of the extensive measurements and the standardization at the basis of our analysis we were generally able to obtain statistical significance with a relatively limited amount of biological replica across all the different treatment used. Therefore, we would like to respectfully disagree with the statement of the reviewer concerning the spreading of the data. Indeed, in light of the relative consistency of our data we were not confronted with many outliers and importantly the rare data sets that distanced themselves from the average, see for example the effect of EGF on cilia length and thickness in figure 2C, D, did so following the general trend of the variation and there was no need of removing them to obtain statistical significance. We would like to apologize for the lack of the information mentioned by the reviewer. As detailed below, we have now introduced several editorial changes to the result and method section to make these points clear according to the reviewer's suggestion.

According to the methods, each data point on the bars in the graphs represents one "n", which is from one animal. However, this reviewer would like a description of how many cells/cilia/fields of view were scored in each animal. Furthermore, in the figure legends the authors write "each data point represents summarized data from one animal" - what is meant by "summarized"?

With the term "summarized" we refer to the fact that for each animal cilia were counted across multiple regions of interest to obtain a final measure that would not be affected by regional variation. We have now reworded the correspondent text to make this point more clear. All requested details have been now added to the "Imaging and Analysis" paragraph of

the “Materials and Methods” section of the revised manuscript. We thank the reviewer for pointing out at these shortcomings.

The manuscript was difficult to read and follow at times, made harder by the use of non-scientific language, poor grammar and long sentences.

We apologize for these shortcomings we have now extensively modified several sections of the manuscript to make it easier for the reader to read it and understand our conclusions.

Moreover, the authors make conclusions based on the data from this manuscript coupled with data from a recent previous article of theirs (Baur et al., 2023), however, their recent previous work is not peer reviewed or published, it is in BioRxIV. In addition, this study does not have a series of cumulative evidence supporting their interpretations/conclusions as is typical in research articles. Consequently, the conclusions drawn are not well supported.

As the reviewer rightly points out our work establishes a link between the ciliary phenotype and the change in the proliferation observed upon ablation of GDF15. The reviewer is also right in mentioning that the analysis of the latter, which is a study consisting of seven main figures and several supplementary data, is described in a separate manuscript. Despite sharing the finding that GDF15 affects proliferation in the embryonic and adult germinal niche, the two manuscripts have completely different focuses. The previous study deals with how the change in proliferation regulates the number of ependymal and neural stem cells, and it has been now been published in Stem Cell Reports (Baur et al. 2024). Instead, the current manuscript focuses on the mechanisms underlying the effect of GDF15 on proliferation showing the involvement of ciliogenesis in this process. This involvement is supported by the fact that modifying the activity of ciliary proteins like GDF15 treatment rescues not only ciliogenesis but also proliferation. Therefore, we would like to respectfully disagree with the reviewer’s statement that our conclusions are not well supported.

Figure 1

Can authors provide explanation for the differential localization of the ciliary proteins in Figure 1? Perhaps counting the GFRAL-positive cilia relative to the other ciliary proteins will reveal a

In the figure we use two different ciliary proteins, i.e. ARL13B and ADCY3, to illustrate the fact that GFRAL is expressed in cilia. It is known that in several brain regions the first is particularly expressed in glial cells and the second is expressed in neuronal cells. We have investigated both markers in previous studies we have found in neurogenic regions and especially in the V-SVZ the presence cilia expressing either or both markers. Whereas it is possible that this reflects cilia from progenitors at different stages of differentiation, the reasons underlying the differential expression of the two proteins are not completely understood. For example, they may also reflect differential epitope accessibility in the two cilia subsets and ad hoc studies would be required to further investigate this issue. Concerning the second question, which is unfortunately only to interpret, it may concern the possibility that GFRAL is associated with a specific group of cilia characterized by a particular pattern of marker expression. To such a concern we can reply that our we did not observe a particular segregation between the pattern of ciliary marker expression and GFRAL and that partial heterogenous pattern of expression was also observed for other proteins like for example Prominin-1. Therefore, our present data do not support this hypothesis. However, cilia are relatively difficult to identify and image at the single organelle level, and therefore further study will be necessary to exclude the existence of very subtle associations between a particular pattern of ciliary marker expression and GFRAL ciliary colocalization.

2. Figure 1E is unclear. The authors claim cilia number is at WT levels in the $Gdf^{-/-}+GDF15$, however, the graph (1E) does not appear to reflect this. There is also an internal consistency, as the authors report cilia number is 23.44, but the graph (1E) indicates it is closer to 40. In addition, there is a statistical "*" above the $Gdf^{-/-}$ control bar (bar 3), but it is not clear what that is being compared to.

The original manuscript contained an error regarding the cilia number, which has now been corrected in the revised version of the manuscript. We have now also changed the statistical labelling in all figures to more clearly indicate which bars are being compared. We would like to apologize for these errors.

3. Figure 1F is not labelled in the Figures.

We have updated the figure legends to ensure that all panels are described.

Figure 2

1. How many times were the Western blots performed? How many tissue samples were analysed ie. What is the n? Densitometry is necessary to quantify the reported differences in pERK. At a minimum, tissues samples from 3 animals should be analysed.

Indeed, throughout the manuscript we have used a minimum number of three biological replica in our quantitative analysis, as mentioned in the method section. We have also used densitometry to quantify ERK phosphorylation. We would like to thank the reviewer for pointing out that this information was not available in the manuscript and apologize for this omission. In the revised manuscript we have now added the quantification of the western blot in Fig. 2B of the revised manuscript.

2. The wild-type and $Gdf15^{-/-}$ control

We are unsure how to interpret this comment. If the reviewer refers to the use of control for samples obtained from the different genotypes, this refers to the untreated, i.e. not exposed to exogenous growth factors or pharmacological agent, samples. We have now made this point clear in the figure legend.

3. It is unclear how the authors can confidently identify the cilia axoneme "length" versus "thickness" in the samples with stunted cilia ie. *actD*, *GDF15 U0126* etc. Labelling the basal body to provide cilia orientation would make the data more compelling.

As the reviewer rightly mentioned we assessed cilia morphology, i.e. length and thickness, by focusing on the ciliary portion projecting out of the cells and we did not use basal body labelling to identify the entire cilia extension. The analysis of cilia morphology especially in primary immunostained tissue is essentially complicated by possible changes in orientation of the organelle that may induce a bias in the organelle measured, i.e. only a subset with a particular orientation, as well as an artifact due to the fact that orientation may affect observed length in maximal intensity projection images. A clear demonstration of the bias introduced by maximal projection analysis is the difficulty for us to find representative pictures consistent with the quantifications, as the reviewer mentions in several comments below. Aware of these challenges we have used only partial z-projections of 2-3 z-planes in our analyses of single cilia, which were reconstructed on the consecutive partial z-projections. As mentioned above, to avoid human bias we reconstruct the recognizable cilia in each region of interest for multiple regions of interest identified in a standardized manner across the apical surface of the germinal niche using two different ciliary markers. Importantly, we did not use tissue section but whole mount preparation to avoid the effect of random sectioning on perceived cilia orientation. As mentioned in our reply above, this

allowed us to reliably and readily detect differences in cilia morphology throughout the different conditions. Since the scope of our study is to determine differences in relative length and not the absolute length of each cilium, we feel that the repeating the measuring to including the analysis of the ciliary base will unlikely bring new conceptual insights. This would only be the case if the mutation and treatments would affect rather than cilia length, the position of the ciliary base with respect to the cell surface. However, this is extremely unlikely as the analysis of the position of the centrosomes in the apical germinal tissue show no major effect of the genotype and scanning electron microscopy reveals no particular invagination of the ciliary pocket in the mutant tissue. Indeed, these experiments reveal a cilia length very similar to the one measured upon immunostaining and imaging, confirming the accuracy of our approach. To further strengthen this aspect, we have now added an image showing γ -tubulin-staining to mark the basal body of the cilium to supplementary Figure S1B. This image shows that a similar position of the ciliary base (γ -tubulin immunopositive shown in gray) across short and long cilia (Arl13b immunopositive shown in red) in both genotypes.

4. The quantification in 2C/D is inconsistent with the corresponding images presented in 1B and 2B. For example, the EGF and actD images in 2B shows elongated cilia and stunted cilia, respectively, however, the spread of the data points/quantification do not reflect this. Furthermore, why wasn't cilia number scored for the 2B images, particularly as some of the treatments appear to show a reduction in cilia number compared to the Gdf^{-/-} in 1B? In addition, as EGF induces pERK, perhaps treatment of GDF^{-/-} with EGF would rescue or partially rescue the ^{-/-} phenotype.

Indeed, although as shown in the quantification, the average length of the cilia is reduced in the embryonic GE lacking GDF15 and treated with exogenous EGFR or with GDF15 in the presence of MAPK inhibitor, some individual cilia are longer, which possibly reflects variable levels of GFRAL expression among cilia. There is also some regional variability possibly among the tissue in terms of abundance of these longer cilia and their number, underscoring the importance of our standardized analysis of multiple regions across the apical surface of the GE. However, the biggest additional problem in selecting representative pictures of cilia length is the bias introduced by the maximal projections images as we discussed in our reply to figure 2 point 3 above. To minimize this interference, we have now modified the panels of the figure and all the figures mentioned below. These new panels illustrate smaller regions of the tissue and are represented by higher magnification images. In addition, we have discussed this issue in the relevant method section and figure legends. We have not followed up the number of primary cilia, because first of all their number was not decreased by application of exogenous GDF15. Moreover, as mentioned above their number was very variable among different preparations, and it was not affected by the treatment, including EGF. Since it displayed a too low resolution for us to be able to use to effectively monitor changes in ciliogenesis and it was not significantly affected by GDF15 treatment, this parameter was not analyzed further. Notably, this variability was not observed with respect to cilia morphology, i.e. length and thickness, we can conclusively say that, in contrast to GDF15 treatment, these parameters are not affected by treatment with exogenous EGF. We have now corrected and clarified this issue in the relevant section of the “Results section” of the revised manuscript.

5. The authors report their previous studies have identified reduced membrane EGFR in Gdf15^{-/-}, therefore, is EGF an appropriate agonist if the receptor levels are reduced in the ^{-/-}? Moreover, the previous studies the authors reference are seemingly not published, the work is reported in BIORXIV

In the current manuscript and in the paper the reviewer is referring to (now published), we show that, independent of the genotype, neural progenitors express EGFR and respond to acute exposure to EGF or with GDF15 with a similar phosphorylation of ERK. Therefore, we feel that the question of whether the two growth factors similarly affect cilia length and proliferation is not only legitimate but also an important one in our effort to understand the importance of the subcellular localization of GFRAL, since unlike the latter, EGFR is not localized in cilia and our data showing that EGF does not affect proliferation in a manner similar to GDF15. Finally, as we mentioned in our reply above, after revision, the study illustrating the effect of GDF15 and EGF on proliferation and apical cells number, are visible on the preprint and have now been accepted for publication.

Figure 3

1. The spread of data in the SAG-treated samples in 3A raises concerns about how meaningful and significant the results are. Rather than using whole tissues, can the cell populations be isolated/cultured/treated in vitro?

The spread of the data concerns only the analysis of *Gli1* transcript and the variability is not unusual, since it is inherent to the low expression of the gene and possible experimental variation due to differences in dissection. Despite this, our data highlight very significant differences among the genotype and of the effect of the treatment, which reflects the expected activation of SHH signalling. Therefore, although as the reviewer mentions, culturing may reduce the spread, we feel that the introduction of an additional experimental system and more artificial conditions would not add to the strength of our conclusions.

*2. Although the SAG-induced expression of *Gli1* is reportedly reduced in the *Gdf*^{-/-} compared to WT, the relative increase in *Gli1* in the SAG-treated *GDF*^{-/-} compared to the DMSO treated *Gdf*^{-/-}, appears similar to the WT ^{-/+} SAG treatment. Therefore, the responsiveness of *GDF*^{-/-} cells to SAG, may be similar to the WT.*

We agree with the reviewer. Indeed, in our study we comment “...smoothened agonist (SAG), a SHH pathway activator, led to a similar significant increase in *Gli1* transcripts in both tissues, indicating that changes in endogenous SHH activation are not dependent on cilia morphology, and that despite bearing shorter cilia, progenitors in the GE are still responsive to SMO relocation.”

*3. What was the rationale behind investigating whether SAG treatment (*Shh* signaling) impacted cilia length or thickness in the ^{-/-}? Did the WT show any changes in cilia dimensions with SAG treatment? If *Gdf15* acts downstream of SAG (Smoothened) to modulate cilia dimensions, it would not be evident in the assay the authors have done. The authors have over interpreted this data.*

In the literature, albeit not in neuronal progenitors, several papers report the effect of SHH on cilia length. Since we found that endogenous SHH is decreased in mutant progenitors we investigated the hypothesis that the decrease in SHH endogenous signalling in mutant precursors may directly be responsible for the effect of the change in cilia morphology. The analysis of the effect of SAG in WT cilia is reported in figure 3 clearly show that unlike GDF15, SAG did not affect cilia morphology despite inducing increase in *Gli1* expression. Since the experiments in questions showed no effect of SAG in cilia morphology, we can conclude that altered SHH signalling is not involved in the effect of GDF15 on cilia morphology. Because we found no effect of SAG on cilia morphology, we also did not investigate whether GDF15 does so downstream of SHH signalling. Notably, in figure 6G, we also show that *Gli1* expression is not altered by the treatment with GDF15, which clearly show that a change in endogenous SHH signalling is not required for GDF15 to be able to

affect cilia morphology. We have now changed several parts of the manuscript, including both results and discussion section, to make this point clearer.

Figure 4

1. It is unclear which image shows the increased ADCY3 immunoreactivity - there is no figure citation. If the authors claim an increase in immunoreactivity, it should be quantified.

2. The authors claim the increase in ADCY3 immunoreactivity was "no longer detected upon application of exogenous GDF15, indicating that the increased in ADCY3 immunoreactivity in ADCY3 in the GE may reflect the shortening of primary cilia." Can authors explain this claim? What is meant by "reflect". Moreover, the apparent ADCY3 immunoreactivity results correlate with the claimed mRNA transcript levels of ADCY3 before and after GDF15 treatment. Therefore isn't that the most likely explanation?

We would like to reply to these comments together, as they are both concerned with the question of whether, besides affecting transcript levels, lack of GDF15 also affects the expression of ADCY3 protein. Besides the qualitative evidence, we have now provided quantitative measurements illustrating that the levels of ADCY3 are increased in mutant cilia in the SVZ, but not in the hippocampus (see panel 4D in the revised manuscript). However, since the commercial antibodies tested in our hand do not recognize the antigen in western blot, we are unable to determine the amount of total protein in the tissues. Notably, the increase in ADCY3 expression was observed both in the fraction of mutant longer and shorter cilia (Cilia length above 3 μm : fluorescence levels: WT 41.94 ± 3.48 vs Gdf15^{-/-} 79.98 ± 2.86 ; cilia length below 3 μm : fluorescence levels: WT 55.57 ± 4.14 vs Gdf15^{-/-} 88.40 ± 2.76 ; n \geq 14; P value $<$ 0.0001; 2-way ANOVA). Therefore, we have now changed the text accordingly in the relevant result section.

3. The rescue of the cilia phenotype using NKY80 or Tubastatin A is not convincing based on the images presented in 4D. Moreover, the images and the corresponding graphs are inconsistent with one another.

We have already discussed at length the problems associated with selecting representative pictures (see also our reply to figure 2 points 3 and 4 above). Also, in this case we have now modified the graph displaying cilia length and thickness to make it easier to relate the quantification to the displayed images.

4. Please clarify the conclusion on line 330/331, what "signaling molecules" are being referred to?

With this sentence we refer to HDAC and ADY3. We have now made this reference explicit in the revised manuscript.

Figure 5B

1. Was the densitometry measured relative to a loading control?

The densitometry for acetylated tubulin was normalized to actin as loading control, as displayed in the western blot image. This has now been noted in the figure legend.

2. The authors seem to propose HDAC6 may more active at cilia in the Gdf^{-/-}, however, all of the analysis is at the population level ie. Western blot and immunofluorescence of whole tissue. Moreover, the HDAC6 tissue staining was not co-localised with a cilia marker. It is unclear how the author expect to identify local changes at cilia using these approaches.

As the reviewer rightly mentions we did not make this hypothesis based on the analysis of the localization of HDAC6. We rather inferred that the enzyme is differentially localized/active

in WT and mutant progenitors by observing the differential effects that a general HDAC6 blocker has on tubulin acetylation and cilia length. According to the current knowledge, HDAC6 contributes to cilia shortening and shedding necessary preceding cell division. Therefore, blockade of HDAC6 should lead to an increase of ciliary length, independent of the genotype, as observed upon blockade of ADCY3, which promotes cilia length in both progenitor groups. Since we see this expected effect only in mutant progenitors but not in the WT, we conclude that this mechanism of cilia length regulation is more active in the first than in the latter progenitor group. We have now modified the text to make clear that our conclusion is based on the speculative analysis of our data.

Figure 6

1. In the results section relating to Figure 6, the figure citations direct the reader to Figure 4 in lines 374-376

We thank the reviewer for pointing out this mistake, which has been now corrected in the revised manuscript.

2. The authors claim to examine a "causal" relationship between cilia and proliferation by treating tissues with inhibitors and seeing if it leads to a decrease in "cell cycle speed" via Ki67 staining. (line 369-374). This approach does not evaluate a "casual" relationship - it is correlative. To demonstrate if it is causal, the phenotype needs to be rescued by modulating the candidate signaling pathway. Moreover, cell cycle speed can not be accurately evaluated via ki67 staining. Were the cells/tissues synchronized before commencing the treatments? Can the cells be isolated, cultured in vitro and the assay performed?

We agree with the reviewer that establishing the speed of the cell cycle for any given population requires not only Ki67 analysis and should be done after synchronization of cell cycle. However, this type of analysis cannot be readily applied to our system which involves the analysis of the proliferation of a highly heterogenous population of progenitors which remain difficult to identify. We also agree with the reviewer that our experiments do not allow to establish a causal relationship, because the manipulation of ADY3 and HDAC6 may affect other cell compartments beside cilia. We apologize for the incorrect use of the terminology and we thank the reviewer for highlighting these overinterpretation of our results. We have now modified the text in the relevant Result section of the manuscript to provide a better description of our data.

3. Why are there so few Ki67-positive cells in the tissue? It appears there are far more ciliated cells than proliferating cells.

We would like to remind the reviewer that the images refer just to the most apical portion of the GE, at an age when many have already entered a quiescent state as stem cells or are undergoing differentiation to become ependymal cells. Several previous studies have shown a low number of Ki67+ cells at this age. Please see as an example the following study (Wang et al. 2009).

Reviewer #2 (Comments to the Authors (Required)):

Baur et al. demonstrated that Growth/differentiation factor 15 (GDF15) regulates the morphology of primary cilia in apical progenitors in the subventricular zone (SVZ). The authors also indicated that the GDF15-HDAC6 axis plays a critical role in the regulation of primary cilia morphology, affects sonic hedgehog (Shh) signaling and the proliferation of apical progenitors. This paper shows an interesting aspect of neural stem cell regulation by primary cilia. However, I find the manuscript somewhat immature, and it was difficult for me to read it. I think the authors should rework the text and reorganize the figures.

Major points.

1. It is very difficult for me to read the manuscript. Each sentence is too long and contains many messages. Please rewrite the results section and make it easy to read.

We apologize for the complexity of the presentation of our analysis, a problem that was highlighted by both reviewers. We have now introduced several editorial changes throughout the manuscript in an effort to improve its clarity.

2. NKY80 is used as 200 μ M in the study. The IC₅₀ of NKY80 for adenylyl cyclase (AC3) is about 132 μ M. I feel very uncomfortable with this compound because of the high dosage. Moreover, the IC₅₀ of NKY80 for AC5 is 8.3 μ M, and some researchers suggest that AC5 and AC6 are also localized in the cilia. Please confirm that the effect of NKY80 is mediated by AC3, but not any other ACs.

As the reviewer rightly mentions, the IC₅₀ of NKY80 for AC3 is 132 μ M; since we attempted a complete instead of just partial inhibition of AC3 activity, 200 μ M is a reasonable concentration for this experiment. The reviewer is also correct regarding the fact that NKY80 is more specific to AC5 and AC6, both of which are also expressed in cilia. We can therefore not exclude the involvement of these Adenylate Cyclases in this process. In this study however, we focused on AC3 due to the availability of specific antibodies and the vaster literature concerning this AC. We can furthermore conclude that the inhibition of non-ciliary ACs, such as AC2 (IC₅₀ 1700 μ M) is not involved in this process and that NKY80 inhibits only ciliary ACs. We have now discussed this issue in more detail in the discussion section of the paper (p. 15, lines 22-31).

3. The order of the figures in the text is difficult to follow. For example, I can't find Figure 1B in the text. Please check the orders in which the figures are listed.

We would like to thank the reviewer for pointing out this shortcoming. In the case of the missing citation, we refer in the text to the quantification, panel 1C but not to the representative images. We apologize for any ensuing confusion. We have now addressed this issue and controlled that every figure panel is mentioned in the study.

4. It is difficult to understand the high mag images. For example, Fig 1A is an image of the germinal eminence and hippocampus at E18. However, I have no idea where the images were taken from. Please clarify the location by showing low mag images or cartoon.

We have now added a graphical representation of the regions the images were taken to Fig. 1A. All subsequent GE and hippocampus images were taken in the same manner. This is also described in the “Imaging and analysis” section of the Materials and Methods.

5. I am very surprised that not all Arl13B positive cilia have ADCY3 in Figure 1A. Did the authors confirm this observation with another ADCY3 antibody? Have the authors used Rabbit Polyclonal Antibody to Adenylate Cyclase III (EnCor Biotechnology Inc, Cat# RPCA-ACIII)?

The fact that ADCY3 is not expressed in all primary cilia in a non-neuronal cell population is not surprising, as its expression is usually consistently seen in primary cilia of neuronal but not glial cell population, see also our reply to comment Figure1 point 1 of the first reviewer. We have tested in the course of the study several antibodies: apart from the ThermoFisher PA5-35382 shown in this paper, we have also used the widely used Santa Cruz sc-588 antibody before it was discontinued, as well as Abcam ab199457; the latter was not used further due to additional unspecific staining in the nucleus. However, ciliary staining was similar with all antibodies with respect to the fact that not all Arl13b⁺ cilia display ADCY3

immunoreactivity, which is consistent with previous independent observations (Antal et al. 2017).

6. It is difficult to interpret the images of primary cilia in Figure 1B. The number of primary cilia in GDF15 KO mice is about 5 times higher than the control images. However, Figure 1E shows that the number is less than 2-fold increase. Please clarify this discrepancy. In Figure 1E, the addition of GDF15 did not change the number of GDF15 KO cilia. However, the number of cilia from GDF15 KO+GDF15 is less than GDF15 KO cilia in Figure 1B.

We would like to thank the reviewer for pointing out this incongruity in the representation of our data. A similar issue concerning the apparent length of the cilia was also raised by the first reviewer (see also our reply to the comments Figure 2 point 3 and 4 of the first reviewer). With respect to the image mentioned by the reviewer it should represent three types of changes, i.e. cilia length, thickness and number. Taken into account the regional variation we have implemented a highly standardized analysis, which included the quantification of cilia properties by identification with at least three ciliary markers and with additional electron scanning microscopy. Whereas this extensive analysis allowed us to readily and consistently detect changes in cilia morphology between genotypes and treatment, the same was not consistently observed with cilia number: In particular, although they were generally higher in the WT than in the mutant tissue, there was a considerable variation among the different preparation in absolute numbers. For this reason and due to the fact that acute GDF15 application did not affect cilia number we did not further pursue this analysis throughout the study. We have now added this information to the relative sections of the revised manuscript and changed the figure panels in an attempt to provide images more representative of the quantitative analysis.

7. I am concerned about the picture of thick cilia in GDF15 KO mice in Figure 1B. I understand that the authors showed SEM images to clarify the thickness. Please include the SEM images in Figure 1. Please also include the high mag image of cilia. Please clearly show the picture of cilia of GDF15 KO mice is not over saturated.

We agree with the reviewer that including in the same figure the results of two different types of analysis allows us to highlight all the different parameters that have to be taken into consideration in such an analysis and therefore it will strengthen our finding that GDF15 ablation affects cilia morphology. Therefore, we have implemented this suggestion. Concerning the apparent oversaturation of the images illustrating cilia thickness, we would like to respectfully point out that we used the same imaging settings, which avoided oversaturation during imaging, of wild type and mutant tissue. Thus, the apparent saturation reflects the difference between the two tissue groups. Please see also our reply to comment 13 regarding the apparent oversaturation.

8. Since the authors showed 4 graphs in Figure 1C-E, please show each pictures in Figure 1B.

We have now added to the figure a representative picture of the WT tissue upon treatment with GDF15.

9. How do you know that the cilia you counted are bona fide primary cilia in Figure 1F and 1G? Please clearly show that these are primary cilia, and not cilia from developing ependymal cells. I think the authors can co-stain with GFAP and beta-catenin to show ependymal cells and type B cells. I think the authors can co-stain with CD133, as shown in the previous paper (Khatri et al., Scientific reports 2014)

Unfortunately, GFAP will be expressed only postnatally and as we have shown in the study mentioned by the reviewer, CD133 is expressed both in ependymal and primary cilia. However, ADCY3 is not expressed in ependymal motile cilia and at E18, the age at which the investigation, multiciliated cells are virtually absent as the differentiation of the ependymal layer occurs postnatally. Moreover, when they do occur, they are easily recognizable and were not included in our analysis. We have now included a representative picture of such a rare ependymal cell to illustrate the visual difference between ependymal and primary cilia (supplementary Fig. S1C). Therefore, it is very unlikely that motile cilia were included in our analysis.

10. Please include pictures of control and GDF15 in Figure 2B, as shown in Figure 2C and 2D.

We have now modified the figure according to the reviewer's indication.

11. Figure 3A is difficult to interpret. I think the "+" indicates the difference between DMSO treatment in GE and SAG treatment in GE. Please show a bar to clearly indicate which graphs you have analyzed for statistics. Please also include "ns (not significant)" for important bar graph comparisons, such as SAG treatment in WT GE and SAG treatment in GDF15 KO GE. Please do the same for Figures 4A, 4E, 4F, 6B and 6C

We have modified the figures according to the reviewer's suggestion.

12. Please include bar graphs of WT DMSO and GDF15 KO DMSO in Figure 3C and 3D.

We have modified the figures according to the reviewer's suggestion.

13. It is difficult to interpret Figure 4C. The WT picture clearly shows ADCY3 localization in most of the cilia in contrast to Figure 1A. I agree that GDF15 KO cilia have increased levels of ADCY3. I think the authors need to quantify the intensity of ADCY3 in Figure 4C. The picture of GDF15 also makes me wonder if the thickness is really different as compared to the control, because it looks like some cilia are oversaturated. As you know, oversaturation affects the thickness of the cilia.

A similar issue was also raised by the first referee, please see also our reply to the comment figure 4 points 1,2 of the first reviewer. The brightness and contrast of the images were increased for better visibility, which may give the impression of high intensity staining. We have now quantified the intensity of ADCY3 expression the data are illustrated in figure 4B of the revised manuscript. When measuring the mean values of ADCY3 and Arl13b in cilia, we found values between 50-100 for ADCY3 (Fig. 4B) and 40-60 for Arl13b (data not shown), with the maximum values being 181 and 98, respectively. These values are well below the 8-bit maximum value of 255 which would indicate oversaturation. Therefore, we are confident that the cilia in these images are not oversaturated and we are seeing real effects on the ciliary morphology.

14. In Figure 5C, it is difficult to interpret the HDAC6 staining. Are these signals real? The authors can use antigen retrieval protocol for immunohistochemistry to obtain better signals.

We would like to point out that, besides a series of negative controls, we have used several antibodies, staining protocols to optimize the protocol for HDAC6 immunostaining. Therefore, we have no reason to believe that the signal observed does not describe the distribution of HDAC6. However, we agree that the data on itself would be of difficult interpretation, because the main change that we see in HDAC6 expression is in the pattern of immunostaining observed upon treatment with tubastatin A. Indeed, the conclusion that we draw on the differences between distribution/activation of HDAC6 between the genotypically

different progenitors are essentially drawn on the analysis of the changes in acetylated tubulin and cilia length (see also our reply to the comment figure 5 point 2 of the first reviewer).

15. Is there a way to show that both NKY80 and Tubastatin A are really working in Figure 4 and 5?

Concerning Tubastatin A, Fig. 5A-D clearly shows a strong increase in acetylated tubulin after Tubastatin A treatment in both western blot and immunofluorescence, which shows that the inhibitor is working. The effectivity of NKY80 was confirmed by cAMP-Glo assay (Promega), which reduced cAMP levels around 80%. We have now included this dataset in supplementary figure S3C to further demonstrate this point.

16. In Figure 6a, I am surprised at the number of Ki67 positive cells you show in the pictures. I thought there should be more proliferating cells in that area. How did you stain the Ki67 antibody? I thought the abcam rabbit Ki67 antibody works best for room temperature overnight condition. The authors can use other Ki67 antibodies, pH3 or PCNA antibodies to confirm the results.

This is also an issue that was raised by the first reviewer, see also our reply to Figure 6 point 2 of the first reviewer. We have taken advantage of Ki67 immunostaining for the analysis of the neural proliferation. In particular the differences between the proliferation of apical progenitors in the WT and mutant GE have been extensively analysed with Ki67 and several other markers of proliferation in a study that was recently accepted for publication. The images shown in this figure refer just to the most apical portion of the GE, at an age when many have already entered a quiescent state as stem cells or are undergoing differentiation to become ependymal cells. Several previous studies have shown a low number of Ki67⁺ cells at this age. Please see as an example the following study (Wang et al. 2009).

17. In Figure 6A, please include pictures of GDF15 treatment, as the bar graphs are shown in Figure 6B and 6C.

We have now modified the figure according the reviewer's suggestion.

18. In Figure 6B and C, please include DMSO treatment graphs as bar graphs.

See Answer to number 21.

19. In Figure 6C, the authors may use pH3 staining to show the mitotic cells,

As mentioned above, in a recent study we have used also PHH3 to investigate differences in mitosis between WT and mutant apical progenitors. In this study we show that the main difference in mitosis between WT and mutant cells concern the number of cells at late stages of mitosis and not total PHH3 cell number. Therefore, we use this type of analysis here to investigate the number of late mitotic cells.

20. In Figure 6D, please include DMSO treated pictures.

There is a DMSO-treated image in Fig. 6A, which shows the same staining as Fig. 6D. We therefore refrained from adding another control image of the same staining and region. We have now indicated this in the Figure legend.

21. In Figures 6E, 6F, 6G and 6H, please include DMSO-treated bar graphs.

We have now modified all figures according the reviewer's suggestion, except Fig. 6H, where the data for the different treatments was collected over separate experiments, where for each animal, the GE of one hemisphere was treated and the other used as control (alternating treatment between left and right hemisphere to avoid bias); the amount of prominin-1-

expressing cells for this control was set to 100% and directly compared to the treated hemisphere of the same animal. Therefore, the controls of the different treatments/animal samples cannot be represented by a common bar, and the individual controls are rather displayed by a line indicating 100%. We hope this explanation alleviates the reviewer's concerns.

22. In Figure 6H, please clarify the meaning of CD133+. Is this due to decreased number of apical progenitors, or apical progenitors without CD133 in the cilia? Some receptors may exit cilia after signal activation. Please include raw data of flow cytometry data of CD133 in the Supp figures. I would like to see all gating, such as FSC, SSC, DAPI and CD133. Please include isotype control and make a tight gating to clearly show that the CD133 signal is real. We would like to clarify this issue. Prominin-1 (CD133) is a marker mostly expressed by progenitors displaying an apical membrane and in the primary cilium. In our previous study we show that GDF15 ablation leads to an increase in the proliferation of apical progenitors, which in turn affects total number of generated apical neural stem cells and ependymal cells. Taking advantage of flow cytometry, we have measured also the total number of apical Prominin-1+ cells using flow cytometry and shown that, as expected, they are increased in the mutant tissue and that this increase is no longer observed upon exposure of the mutant tissue to exogenous GDF15. Since we here found that Tubastatin A but not SAG affects the total number of proliferating apical progenitors we investigated whether the two modulators also differ with respect to their effect on the number of total apical cells. Consistent with the differences observed in terms of proliferation we found that the first but not the latter, reproduces the effect of GDF15 also with respect of the number of apical progenitors. We apologize for the lack of clarity; we have now modified our result section to make this point clearer. We have also added example of FACS plots illustrating the setting used for the analysis of apical cells in supplementary Fig. S4.

Minor points.

1. Please include catalogue numbers in the method section.

We have implemented this change in the revised manuscript.

2. Please include the molecular size in the figure of western blot pictures.

We have now modified the figure according the reviewer's suggestion.

References:

- Antal, M. C., K. Benardais, B. Samama, C. Auger, V. Schini-Kerth, S. Ghandour, and N. Boehm. 2017. 'Adenylate Cyclase Type III Is Not a Ubiquitous Marker for All Primary Cilia during Development', *PLoS One*, 12: e0170756.
- Baur, K., C. Carrillo-Garcia, S. San, M. von Hahn, J. Strelau, G. Holzl-Wenig, C. Mandl, and F. Ciccolini. 2024. 'Growth/differentiation factor 15 controls ependymal and stem cell number in the V-SVZ', *Stem Cell Reports*, 19: 351-65.
- Wang, B., R. R. Waclaw, Z. J. Allen, 2nd, F. Guillemot, and K. Campbell. 2009. 'Ascl1 is a required downstream effector of Gsx gene function in the embryonic mouse telencephalon', *Neural Dev*, 4: 5.

April 19, 2024

RE: Life Science Alliance Manuscript #LSA-2023-02384-TR

Dr. Francesca Ciccolini
Heidelberg University
Department of Neurobiology, Interdisciplinary Center for Neurosciences (IZN), University of Heidelberg
INF366
Heidelberg 69120
Germany

Dear Dr. Ciccolini,

Thank you for submitting your revised manuscript entitled "GDF15 controls primary cilia in the V-SVZ thereby affecting progenitor proliferation.". We would be happy to publish your paper in Life Science Alliance pending final revisions necessary to meet our formatting guidelines.

- please be sure that the authorship listing and order is correct
- please upload all figure files as individual ones, including the supplementary figure files; all figure legends should only appear in the main manuscript file
- please add a Category and Keywords for your manuscript in our system
- please add the Twitter handle of your host institute/organization as well as your own or/and one of the authors in our system
- please note that the titles in the system and manuscript file must match
- please consult our manuscript preparation guidelines <https://www.life-science-alliance.org/manuscript-prep> and make sure your manuscript sections are in the correct order
- please add your main, supplementary figure, and table legends to the main manuscript text after the references section
- please use the [10 author names, et al.] format in your references (i.e. limit the author names to the first 10)
- please add callouts for Figure S4A-D to your main manuscript text

A. FINAL FILES:

B. MANUSCRIPT ORGANIZATION AND FORMATTING:

Sincerely,

April 26, 2024

RE: Life Science Alliance Manuscript #LSA-2023-02384-TRR

Dr. Francesca Ciccolini
Heidelberg University
Department of Neurobiology, Interdisciplinary Center for Neurosciences (IZN), University of Heidelberg
INF366
Heidelberg 69120
Germany

Dear Dr. Ciccolini,

Thank you for submitting your Research Article entitled "GDF15 controls primary cilia morphology and function thereby affecting progenitor proliferation". It is a pleasure to let you know that your manuscript is now accepted for publication in Life Science Alliance. Congratulations on this interesting work.

DISTRIBUTION OF MATERIALS:

Again, congratulations on a very nice paper. I hope you found the review process to be constructive and are pleased with how the manuscript was handled editorially. We look forward to future exciting submissions from your lab.

Sincerely,
